# Edible mycelium bioengineered for enhanced nutritional value and sensory appeal using a modular synthetic biology toolkit

Vayu Maini Rekdal[1,2,3], Casper R. B. van der Luijt [3,4,5,6], Yan Chen[3,6], Ramu Kakumanu[3,6], Edward E. K. Baidoo[3,6], Christopher J. Petzold [3,6], Pablo Cruz-Morales[4] & Jay D. Keasling [1,3,4,6,7,8] ✉

Filamentous fungi are critical in the transition to a more sustainable food system. While genetic modification of these organisms has promise for enhancing the nutritional value, sensory appeal, and scalability of fungal foods, genetic tools and demonstrated use cases for bioengineered food production by edible strains are lacking. Here, we develop a modular synthetic biology toolkit for *Aspergillus oryzae*, an edible fungus used in fermented foods, protein production, and meat alternatives. Our toolkit includes a CRISPR-Cas9 method for gene integration, neutral loci, and tunable promoters. We use these tools to elevate intracellular levels of the nutraceutical ergothioneine and the flavor-and color molecule heme in the edible biomass. The strain overproducing heme is red in color and is readily formulated into imitation meat patties with minimal processing. These findings highlight the promise of synthetic biology to enhance fungal foods and provide useful genetic tools for applications in food production and beyond.

The global food system has been identified as one of the major contributors to climate change. Food production is responsible for an estimated one-third of global greenhouse gas emissions and contributes to widespread environmental degradation, biodiversity loss, and the emergence of new diseases[1–3]. Transitioning food production away from resource-intensive industrial animal agriculture toward alternative methods, including microbial processes, is critical for mitigating these negative planetary impacts and sustainably feeding a growing global population that is estimated to reach over 9 billion by 2050[1,2,4–7]. Among other applications, microbes can be used for upcycling byproducts[8], as hosts for production of environmentally taxing small molecules and proteins[9,10], and for producing nutritious biomass that can be consumed directly[11]. Compared to animal agriculture, microbial food production can offer increased resource efficiency and safety, more precise control of production, reduced animal suffering, and a reduced environmental footprint[5].

Filamentous fungi, a diverse group of microorganisms that includes molds and mushrooms, have several advantages compared to other hosts for microbially based food production[12] (Fig. 1A). In addition to the historical use of many fungi for safe and delicious fermented foods[13], the naturally high secretory capacity of these organisms makes them powerful hosts for production of proteins for

[1]Department of Bioengineering, University of California, Berkeley, CA 94720, USA. [2]Miller Institute for Basic Research in Science, University of California, Berkeley, CA 94720, USA. [3]Joint BioEnergy Institute, Emeryville, CA 94608, USA. [4]Novo Nordisk Foundation Center for Biosustainability, Technical University of Denmark, 2800 Kgs. Lyngby, Denmark. [5]Department of Food Science, University of Copenhagen, 1958 Frederiksberg, Denmark. [6]Lawrence Berkeley National Laboratory, Biological Systems and Engineering Division, Berkeley, CA 94720, USA. [7]California Institute of Quantitative Biosciences (QB3), University of California, Berkeley, CA 94720, USA. [8]Department of Chemical and Biomolecular Engineering, University of California, Berkeley, CA 94720, USA. ✉e-mail: keasling@berkeley.edu

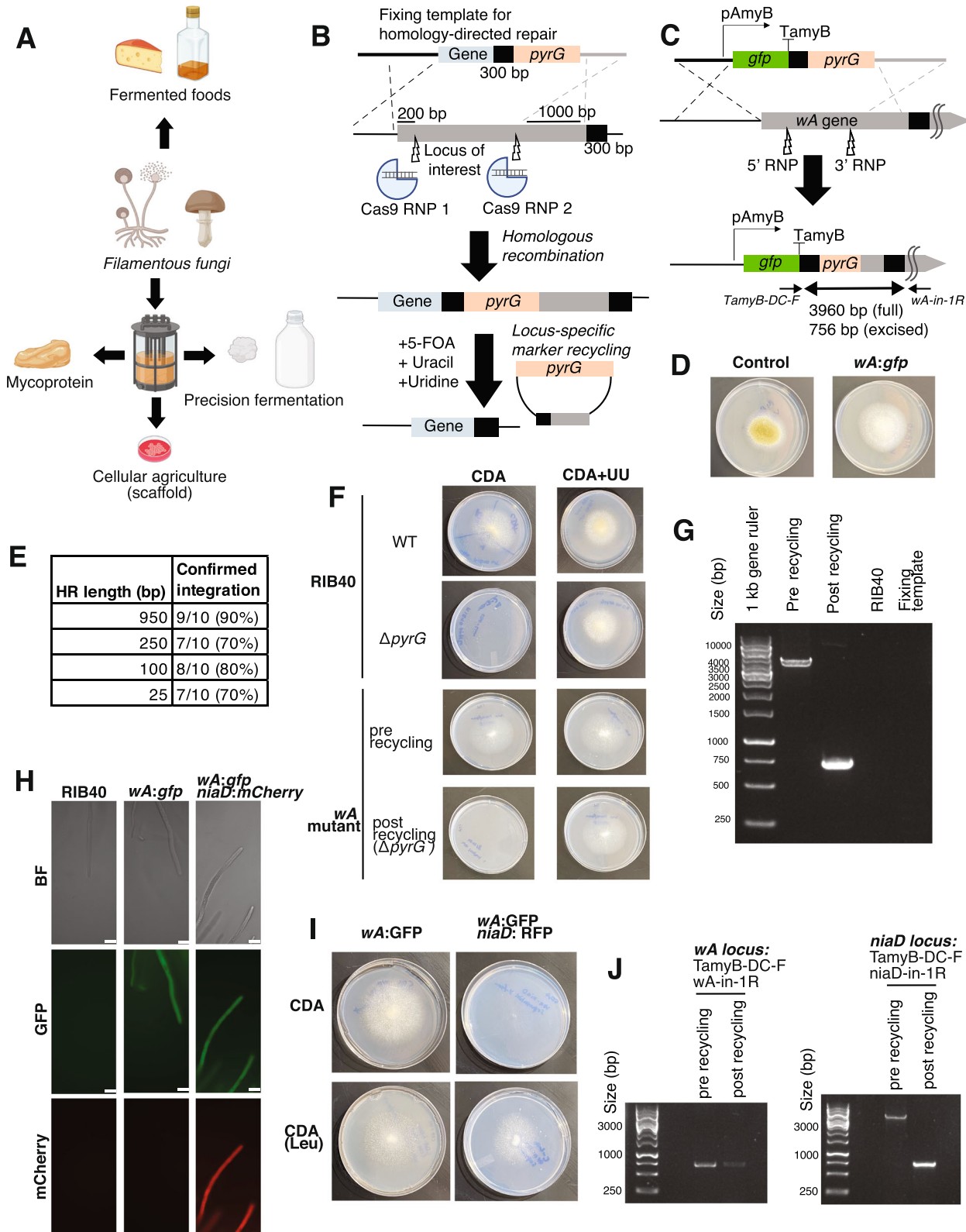

food and other uses[14]. Additionally, many fungi rapidly degrade and grow on complex substrates such as food byproducts or lignocellulose, which can alleviate the cost and environmental burden associated with highly purified substrates such as glucose[9,15]. Finally, owing to its filamentous morphology which mimics the structure of animal muscle, fungal biomass (mycelium) can be formulated into meat alternatives with convincing textures (mycoprotein), and used as

scaffolds for adherent animal cells in cellular agriculture[11]. A recent Life Cycle Assessment revealed that substituting 20% of animal protein with mycoprotein by 2050 could lower methane emissions as well as reduce deforestation and associated $CO_2$ emissions by half, underscoring the concrete environmental benefits of fungal foods[6].

Fungal food production is a rapidly growing area with vast commercial interest and potential, and a growing number of meat

**Fig. 1 | A recyclable RNP-based CRISPR-Cas9 method for efficient gene integration and expression in *A. oryzae*. A** Fungal applications in sustainable food production. The figure was created was created using BioRender (http://BioRender.com). **B** Strategy for RNP-based CRISPR-Cas9 editing. Upon integration at the correct locus, the *pyrG* selection marker becomes flanked by two identical 300-bp sequences. Counter-selection using 5-fluoroorotic acid (5-FOA) allows locus-specific marker excision. **C** Strategy for integration of a GFP-expression cassette (pAmyB promoter) at the *wA* locus, which controls spore pigmentation, in *A. oryzae* RIB40. The two squiggly lines indicate that not the whole 6.7-kb *wA* gene is shown. Primers TamyB-DC-F (on fixing template) and wA-in-1R (on chromosome) were used to confirm successful insertion. Ladder is Generuler 1 kb ladder (Thermo Scientific). **D** The *wA:gfp* transformant has white conidia instead of the yellow-green conidial pigmentation of the RIB40 strain (grown on PDA medium). **E** Efficiency of gene integration at the *wA* locus as a function of the homology arm length in the fixing template. 10 colonies were randomly

selected at each length and subjected to colony PCR. **F** Following *pyrG* marker recycling from the *wA* locus the strain displays uracil/uridine auxotrophy when grown on CDA medium. UU = uracil and uridine supplementation **G** The auxotrophy was accompanied by clear marker loop out from the *wA* locus. Ladder is Generuler 1 kb ladder (Thermo Scientific). **H** The looped-out *wA:gfp* strain (Δ*pyrG*) was transformed with an mCherry expression cassette targeted at the *niaD* locus, which controls nitrate assimilation. Microscopy in the *wA:gfp* and *niaD:mCherry* strain confirmed the expected pattern of protein expression. Scale bar = 25 μm. **I** The resulting transformant only grows with leucine (CDA-Leu) as the nitrogen source instead of nitrate (CDA), as indicated by the colony radiating from the center of the plate. **J** The *wA:gfp* and *niaD:mCherry* strain was subjected to marker recycling from the *niaD* locus. PCR confirmation indicates that marker recycling was successful from the *niaD* locus (see Supplementary Fig. 3 for details), while the *wA* locus remained unchanged. Ladder is Generuler 1 kb ladder (Thermo Scientific).

and dairy substitutes based on fungi are now available on the market across Europe, U.S., and Asia[12,16]. While these products showcase the astonishing versatility and commercial promise of fungi for sustainable food production, most current products are based on a limited group of non-engineered strains, which have inherent limitations in their metabolism, structure, and industrial capacity. Genetic engineering could overcome these limitations and further expand beyond naturally occurring biodiversity, allowing new uses and applications of fungi in human food production[17]. For instance, a synthetic gene expression tool in *Trichoderma reesei*, an industrial fungus traditionally used for enzyme production, recently enabled production of gram-scale quantities of egg white and milk protein[14,18]. However, like many other industrial fungi, *T. reesei* has no history of safe or palatable consumption by humans, which limits the possible food contexts in which the fungus can be utilized, such as the highly efficient and sustainable production of fungal biomass for human food. Extending such synthetic biology tools and approaches to historically consumed, food-safe, edible fungi could expand the engineering possibilities for fungal food production, including enhancing fermented foods and altering the properties of mycoprotein to better suit human dietary needs and preferences. However, synthetic biology tools and demonstrated use cases for bioengineered food production by historically consumed food-safe filamentous fungi are lacking.

Here, we develop a modular synthetic biology toolkit for *Aspergillus oryzae*, an edible fungus with a long history of safe and palatable human consumption, and demonstrate its applicability for enhancing fungal foods. Our toolkit includes a CRISPR-Cas9 method for precise and efficient gene modification, neutral loci for targeted gene insertion, and tunable promoters, including bidirectional promoters as well as a synthetic expression system that offers strong gene expression independent of the medium composition. We use these tools to engineer the nutritional value and sensory appeal of the edible fungal biomass for alternative meat applications. We overproduce ergothioneine, a potent antioxidant, at levels that are higher than in mushrooms, the largest source of this molecule in the human diet. Additionally, we engineer the eight-step heme biosynthetic pathway to create an edible biomass that contains heme at levels approaching those found in leading plant-based meats incorporating heme for flavor and color. In contrast to plant-based protein, the engineered fungal biomass can be readily formulated into meat-like patties without the need for extensive processing, protein purification, or ingredient addition. In addition to demonstrating the potential of bioengineering edible fungi, this work provides synthetic biology tools and approaches that could be useful for fungi across diverse applications and industries.

## Results

### A recyclable CRISPR-Cas9 method for efficient gene integration and expression

We selected *Aspergillus oryzae* (koji mold) as our model edible fungus and engineering target, as this fungus has a long history of safe use and human acceptance in fermented foods[13], has a biomass with a palatable and umami-rich flavor that is commercially available as mycoprotein[19,20], secretes high amounts of protein[21], is used industrially for enzyme production[22], and has demonstrated promise as a scaffold for animal cells in cellular agriculture[23]. To enable synthetic biology efforts across the diverse food applications of this edible fungus, we first set out to create a comprehensive genetic toolkit, including a method for efficient gene integration, neutral loci for high expression, and tunable promoters.

In designing our toolkit, we first considered the challenge of efficiently integrating heterologous genes in desired genomic locations. Filamentous fungi are notoriously poor at homology-based recombination and thus transformation with linear DNA templates often results in off-target, ectopic integrations[24]. To overcome this, strains deficient in non-homologous end-joining (NHEJ) have historically been used[25]. However, disruption of NHEJ presents potential issues with genomic instability or increased risk of DNA damage[26]. Recently, CRISPR-Cas9 has revolutionized the ability to transform and genetically modify fungi, including *A. oryzae*[18,27–29]. For example, the recently developed state-of-the-art CRISPR-Cas9 method for *A. oryzae* allows high-efficiency modification of strains proficient in non-homologous end-joining (NHEJ)[28]. The method uses plasmids to drive constitutive Cas9 and sgRNA expression, and the plasmid can be readily removed using selection, which enables sequential rounds of transformation and genome modification[28].

To efficiently engineer *A. oryzae*, we sought to develop an alternative, easy-to-use CRISPR-Cas9 approach that is compatible with readily available commercial reagents[26], minimizes the possibility of off-target effects and toxicity resulting from constitutive Cas9 expression[30–32] and incorporates a straight-forward phenotypic screen to verify that integration at the locus of interest has taken place[23]. Rather than encoding the Cas9 and sgRNAs from a plasmid, our method involves direct transformation of CRISPR-Cas9 Ribonucleoprotein complexes (RNPs), which can be formed in vitro from commercially available Cas9 protein and sgRNAs. At the start of our study, the RNP-based approach had been demonstrated as a strategy for rapid and precise genome editing compatible with high-throughput screening in diverse filamentous fungi[29,33–36] but had not been experimentally validated for gene integration in *A. oryzae*.

In our method, the DNA template introduced to fix the Double-Stranded Breaks (DSBs) harbors a *pyrG* marker to allow for both positive selection using uracil/uridine auxotroph and negative selection using media with 5-fluoroorotic acid (5-FOA). Although *pyrG*

marker recycling has been demonstrated in *A. oryzae*, previous approaches did not incorporate CRISPR-Cas9 and required an NHEJ-deficient strain to avoid off-target integrations[37]. To overcome potential issues with ectopic integrations in wild-type *A. oryzae*, we designed the system such that a successful loop out of the *pyrG* marker can only occur if the fixing template is integrated at the locus of interest, in which case it will be flanked by two identical 300 bp sequences. In this system, ectopic integrations resulting from NHEJ are unable to loop out and survive on media supplemented with 5-FOA (Fig. 1B). This phenotypic screening approach to locus-specific gene integration and *pyrG* marker recycling was first established using a plasmid-based CRISPR-Cas9 system in the related *A. niger*, where it allowed precise and efficient genome modification[27]. In *A. niger*, all colonies surviving on 5-FOA had the expected genome modification, highlighting the robustness of this approach. The *A. niger* method was used for mutation and gene deletion but was not utilized for integration and expression of proteins[27].

To evaluate our RNP-based method for integration and expression in *A. oryzae*, we first targeted a GFP-expression cassette to the *wA* locus, which controls spore pigmentation, into a Δ*pyrG* mutant of the common laboratory strain RIB40[28] (Supplementary Table 1). This experiment yielded strains displaying the expected white spore phenotype, GFP expression, and fixing template insert (Fig. 1C, D, and Supplementary Fig. 1A, B). To explore whether this method works beyond the model laboratory strain RIB40, we collected a group of *A. oryzae* strains with distinct industrial uses and geographical origins (Supplementary Fig. 2). Whole-genome sequencing of these strains revealed that these strains are phylogenetically distinct from one another and display minor variations in both the number of coding genes and biosynthetic gene clusters (Supplementary Table 2). To enable gene editing, we first generated Δ*pyrG* strains by targeting two RNPs to the *pyrG* locus and plating on agar supplemented with 5-FOA, uracil, and uridine to select mutants. PCR amplification of the region revealed clear mutations at the predicted sgRNA cut sites, including deletions and insertions, likely resulting from erroneous fixing by *A. oryae*. We observed no off-target effect on the surrounding genes despite not providing a fixing template, highlighting the precision of the RNP-based editing method (Supplementary Fig. 3). We then successfully introduced a GFP-expression cassette at the *wA* locus to alter the spore phenotype and establish heterologous protein production across the strain collection (Supplementary Fig. 4). These strains had not previously been genome sequenced or edited, suggesting that "wild" *A. oryzae* strains with potentially favorable phenotypes could be efficiently modified using our method. Moreover, these results indicate that the Δ*pyrG* mutants required for our method are readily generated in a single-step transformation.

The RNP-based CRISPR-Cas9 method was highly efficient. PCR amplification of RIB40 transformants at the *wA* locus revealed a 90% integration efficiency with 950 bp homology arms (Fig. 1E), similar to the high targeting efficiency of the previously developed plasmid-based CRISPR-Cas9 system[28]. Even with homology arms as short as 25 bp, the method proved highly efficient (70%), suggesting that PCR primer overhangs could be used to specify the integration locus of interest (Fig. 1E). The high efficiency with such small homology arms is consistent with findings from other filamentous fungi[34,38]. We also found that we could miniaturize the transformation to smaller volumes and remove the final top agar step without major decreases in integration efficiency, making the transformation process quicker and easier compared to the standard *A. oryzae* protoplast transformation protocol and compatible with a microplate format[28,35] (Supplementary Fig. 1E). Although our method utilizes two RNP complexes for each locus to maximize the likelihood of DSB based on previous successes in fungi[27,39–41], we found no major difference in integration efficiency between using one and two RNP complexes across two distinct genomic loci in *A. oryzae* (*wA* and *niaD*) (Supplementary Table 3).

A core design feature of our method is the ability to recycle the *pyrG* marker upon insertion to the correct locus, as transformants should undergo marker excision in the presence of 5-FOA. We first confirmed successful recycling from the *wA* locus. Consistent with previous results in *A. niger*[27], surviving colonies displayed uracil-uridine auxotrophy and marker excision, indicating successful *pyrG* removal. We observed this across three out of three colonies analyzed, highlighting the robustness of the growth-based method to assess locus-specific marker recycling (Fig. 1F, G, and Supplementary Fig. 1C, D). Finally, we successfully integrated GFP-expression cassettes and excised *pyrG* markers at the *niaD* locus, which controls nitrate assimilation, and the *yA* locus, which contributes to spore coloration (Supplementary Fig. 5 and Supplementary Table 1)[28].

The recyclability of the *pyrG* marker allows for potentially endless rounds of sequential engineering. To evaluate this possibility, we transformed the looped-out *wA:gfp* strains with an *mCherry* cassette targeted to the *niaD* locus. Positive transformants demonstrated the expected phenotype and protein expression (Fig. 1H, I). We then successfully recycled the *pyrG* marker from the *niaD* locus, enabling sequential engineering using our marker recycling approach (Fig. 1J). Finally, we also targeted the *wA* and *niaD* loci simultaneously in a single experiment. However, in contrast to the high efficiency observed with single integration at the *wA* locus, simultaneous modification at the *niaD and wA* loci was only 30% efficient with 950 bp homology arms (Supplementary Fig. 6). This is consistent with previous findings of reduced efficiency with multiple RNP complexes and fixing templates in fungi[35]. Overall, these results establish the RNP-based method as a method for genome modification and protein expression in diverse strains of the edible fungus *A. oryzae*. This method displays a comparable high efficiency and scope as the plasmid-based method for genetically engineering this fungus[28]. The use of commercially available reagents and the ability to phenotypically screen for insertion at the locus of interest makes the protocol easy to use.

## Identification and evaluation of neutral loci for gene expression

After establishing the RNP-based CRISPR-Cas9 method for gene modification, we considered another challenge in genetic engineering of filamentous fungi: where to integrate genes for overexpression. While multi-gene expression has been achieved in *A. oryzae* for natural products biosynthesis, the historically preferred method involves plasmids that integrate randomly throughout the genome[42]. These can cause unintended pleiotropic effects or genomic instability and make it challenging to compare phenotypes between constructs and strains[43]. In contrast, neutral loci, intergenic regions, and genomic safe havens that allow targeted expression without interfering with host physiology, is a standard feature of engineering for many bacteria and yeasts such as *S. cerevisae*[44,45]. Recently, genome sequencing of *A. oryzae* transformed with randomly integrated plasmids revealed two intergenic regions (called "hot-spots") that were successfully targeted with CRISPR-Cas9 for expression of natural products genes[46]. However, to this date, neutral loci have not been systematically identified and evaluated for the efficiency of gene integration and level of protein expression across the *A. oryzae* genome. This information, along with readily available plasmids and DNA parts targeted to characterized loci, is critical to advance engineering efforts, as has been shown in *S. cerevisae*[45].

We took a computational approach to identify candidate-neutral loci in *A. oryzae*. We first identified intergenic regions in the *A. oryzae* RIB40 genome. Using publicly available RNA-sequencing data across diverse conditions and growth stages, we ranked the expression level of the two genes immediately surrounding the intergenic region, thus generating a list of candidate loci predicted to enable high gene expression (Fig. 2A and Supplementary Data file 1). From this set, we selected 10 promising high-expression regions (>4.8 kb) spread across *A. oryzae* chromosomes for further evaluation (Fig. 2B, Supplementary

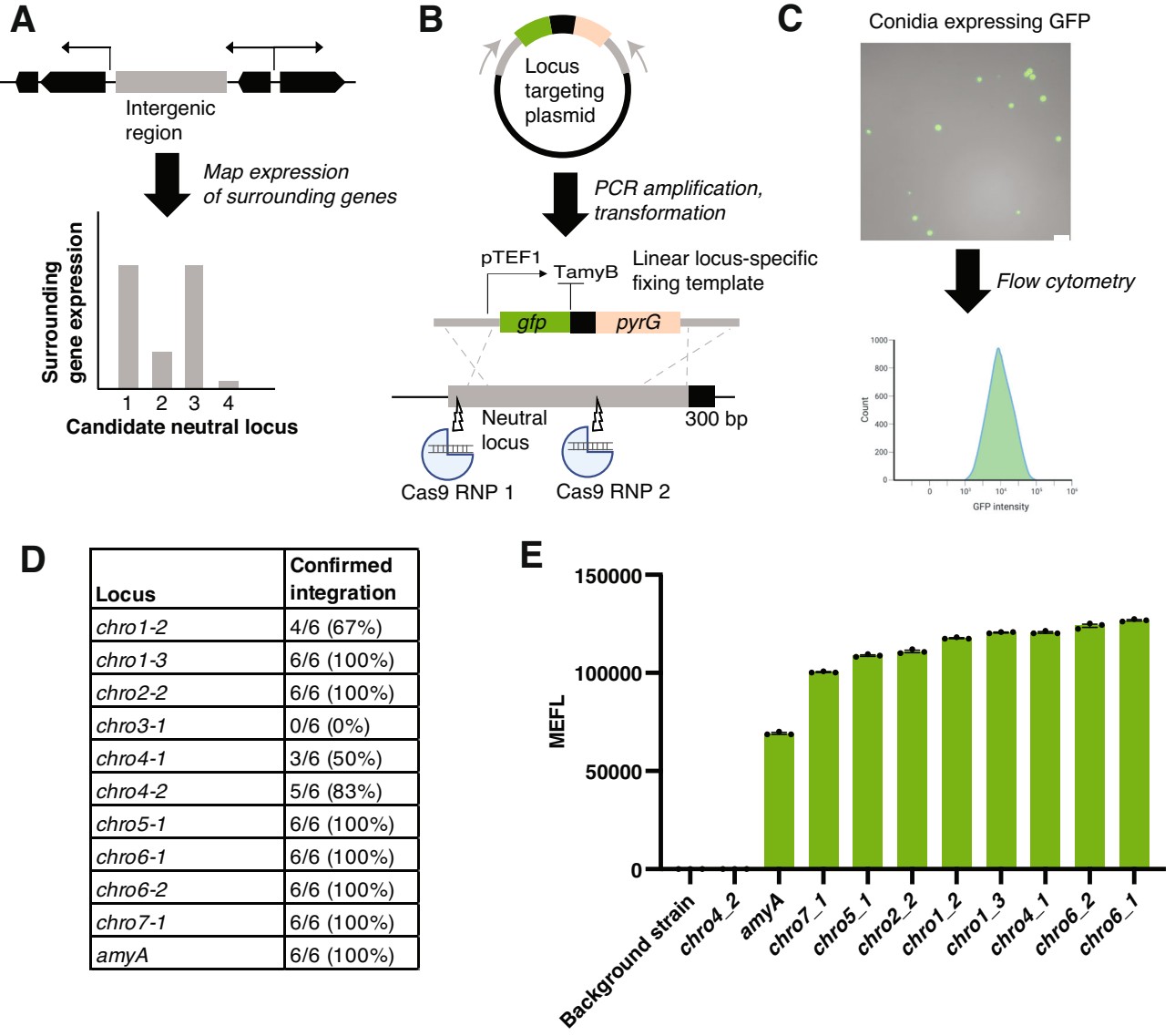

**Fig. 2 | Identification and evaluation of neutral loci for gene expression. A** A computational approach was used to identify intergenic regions with high expression of surrounding genes. The highest expressing regions were selected as promising neutral loci for further experimental evaluation. **B** Targeting plasmids were designed with the 5′ and 3′ homology arms as well as the specific 300 bp sequence for the locus of interest. The plasmid harbors a GFP-expression cassette driven by the constitutive pTEF1 promoter and terminated by the commonly used TamyB terminator, both from *A. oryzae*. The plasmids were cloned in *E. coli* and were linearized using PCR to create linear fixing templates that target the locus of interest. **C** Flow cytometry of conidia constitutively expressing pTEF1 was used to evaluate expression strength at the neutral locus of interest and determine their suitability for engineering efforts. A representative microscopy image showing GFP expression from *A. oryzae* conidia is shown. Scale bar = 25 μm. The flow cytometry figure was created was created using BioRender (http://BioRender.com). **D** Integration efficiency of GFP-expression cassette at neutral loci. All loci except for *chro3_1* displayed a high efficiency of integration (>50%), as assessed by PCR. We could not detect insertion at *chro3_1* by PCR. **E** GFP expression (expressed as Mean Equivalents of Fluorescein, or MEFL) across neutral loci. All loci except for *chro4_2* displayed expression levels above the background strain (RIB40) and were higher than the *amyA* locus, which was included as a positive control to validate the method. Results are average and standard error of the mean (SEM) of three biological replicates.

Tables 4 and 5). We then integrated cassettes harboring GFP under control of the strong, constitutive pTEF1 promoter, and assessed fluorescence using flow cytometry on the conidia of looped-out strains[47] (Fig. 2B, C and Supplementary Fig. 7).

Out of 10 tested loci, 9 showed high-efficiency integration (>50%) (Fig. 2D). We could not detect successful gene insertion at chro3_1 by PCR amplification, suggesting issues with PCR amplification or the integration itself. Following marker loop out, we detected GFP expression above background levels from 8 of the remaining 9 loci, with chro4_2 displaying no clear GFP expression (Fig. 2E). Expression levels were largely consistent across the loci, with the highest (chro6-1) showing ~25% higher expression than the lowest

(chro7-1) (Fig. 2E). All loci displayed higher expression than the *amyA* locus, which was included as a positive control to validate the method. Finally, colony growth and morphology were consistent between all strains and similar to those of the background strain, suggesting no gross effects of gene integration and expression on fungal growth (Supplementary Fig. 8). Overall, these efforts identified not only neutral loci, but also a set of plasmids and sgRNAs that facilitates easy transformation and expression for diverse purposes (Supplementary Table 5). Our computational identification and experimental evaluation using flow cytometry provides a framework for how to identify and evaluate promising candidate loci across fungal hosts.

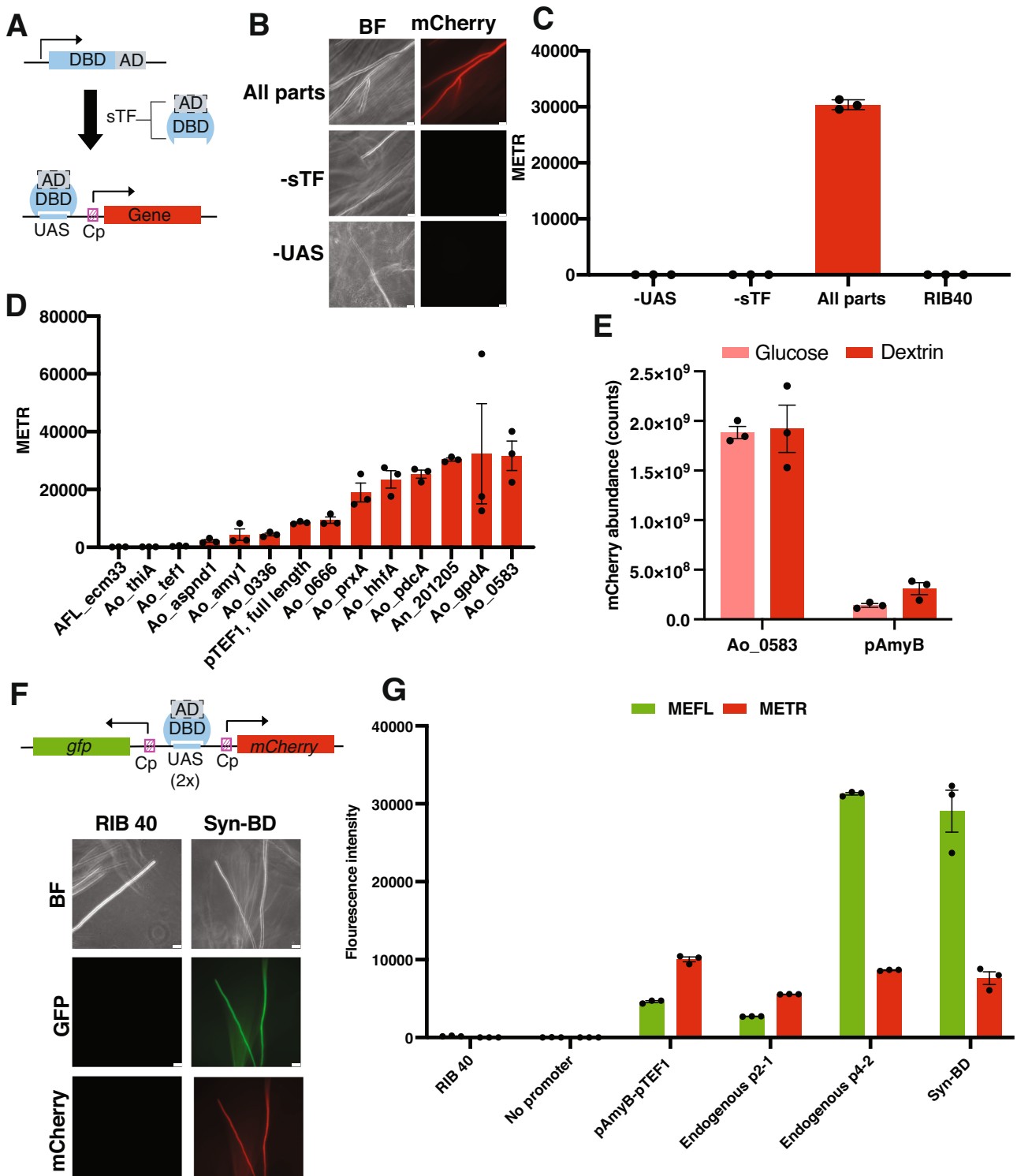

## Expansion of the promoter toolkit using a synthetic expression system and bidirectional promoters

An additional challenge in engineering edible filamentous fungi is the narrow set of characterized parts available for gene regulation, as fungal promoters remain limited in both sophistication and scope. For example, only a handful of endogenous promoters have been used for gene expression in *A. oryzae*, and these are either regulated by the nutrient source (such as the amylase or glucoamylase promoters), have a limited dynamic range, or a poorly understood mode of regulation[48–50]. Synthetic expression systems (SES), which are widely available in yeast and bacteria and increasingly in mammalian cells and plants, could address the technical limitations of current fungal expression tools and expand engineering opportunities in *A. oryzae*[51–57]. SES couple synthetic transcription factors (sTF) with minimal core promoters (Cp) and DNA binding sites (UAS) and offer an orthogonal and highly programmable mode of gene expression[18] (Fig. 3A). The Tet-On SES has shown promise in *A. niger* and *A. fumigatus* for inducible and titratable gene expression[58,59], but food applications are limited by the cost and potential food incompatibility of the small molecule inducer. In contrast, a constitutive SES based on the Bm3R1 DNA binding domain and the VP16 activation domain was recently established in the two non-edible, industrial filamentous fungi

**Fig. 3 | Expanding the promoter toolkit using a synthetic expression system and bidirectional promoters. A** Design of synthetic expression system (SES). Coupling a synthetic transcription factor (sTF, composed of an Activating Domain = AD and DNA binding domain = DBD) and Upstream Activating Sequences (UAS) enables orthogonal and highly programmable gene expression from core promoters (Cp). **B** Confirmation of the Bm3R1-VP16-based SES in *A. oryzae* using the An_201205 core promoter. Fluorescence imaging that the SES in *A. oryzae* requires both the sTF and the UAS for expression. Scale bar = 50 μm for −UAS, 25 μm for other strains. **C** Conidia from strains shown in (**B**) were subjected to flow cytometry for fluorescence quantification of the constitutively expressed mCherry (expressed as Mean Equivalents of Texas Red, or METR). Results are average and SEM of three biological replicates. **D** Core promoter screen using the SES in *A. oryzae*. 200-bp sequences were cloned upstream of mCherry and fluorescence intensity was quantified using flow cytometry of conidia. The full-length, constitutively expressed promoter pTEF1 was included as a benchmark for promoter strength. Results are average and SEM of three biological replicates. **E** Proteomic comparison of intracellular mCherry abundance between the core promoter, Ao_0583, and the full-length starch-inducible endogenous promoter pAmyB from *A. oryzae*. Proteomics was conducted on lyophilized mycelia grown in liquid cultures (CDA medium with dextrin or glucose as the sole carbon source). Results are average and SEM of three biological replicates. **F** A minimal bidirectional promoter (Syn-BD) constructed of 2x UAS binding sites and the *gpdA* and *hhfA* core promoters can drive dual mCherry and GFP expression. Scale bar = 25 μm for RIB40, 50 μm for engineered strain. **G** Identification and evaluation of endogenous bidirectional promoters from *A. oryzae*. Flow cytometry quantification indicated that two promoters, p2-1 and p4-2, could drive bidirectional gene expression at varying levels. p4-2 was similar to Syn-BD in terms of expression. A concatenated sequence of pAmyB and pTEF1 pointing in opposite directions was included as a positive control. MEFL = mean equivalents of fluorescein. METR = mean equivalents of Texas red. Results are average and SEM of three biological replicates.

*A. niger* and *T. reesei*. The highly modular SES showed high programmability and stability across the two hosts and afforded high secreted protein expression independent of the composition of the growth medium[18].

We sought to expand the engineering possibilities in the edible *A. oryzae* by building on these advances in the industrial workhorses *A. niger* and *T. reesei*. To first establish SES as a mode for gene regulation in *A. oryzae*, we initially evaluated the ability of the previously characterized Bm3R1-NLS-VP16 sTF to drive mCherry expression from a core promoter. Using the RNP-based CRISPR-Cas9 integration tools and neutral loci, we genetically integrated the sTF and drove low levels of basal expression of this transcription factor using a characterized core promoter from *A. niger* (An008). In a separate genomic location, we integrated an mCherry cassette harboring 6x UAS upstream of the *A. niger* An201205 core promoter. There was clear expression of mCherry in mycelia and conidia using the full system. The UAS and the sTF were both necessary for activity, validating the predicted function of the SES in *A. oryzae*[18] (Fig. 3B, C).

To explore the programmability of this modular SES in *A. oryzae*, we initially focused on core promoters, as the identity of these short 200-bp sequences influences the level of gene expression upon sTF binding[18,57]. Using available transcriptome data and a curated list of promoters from highly expressed *A. oryzae* and *Aspergillus flavus* genes, we first assembled an initial library of twelve 200-bp core promoters and evaluated their ability to drive mCherry expression in the SES (Supplementary Table 6). We used flow cytometry of conidia[47] as an initial screening approach to assess expression and used the strong, constitutive pTEF1 promoter as a benchmark for comparison. Three of twelve selected core promoters did not drive mCherry expression at levels above the background strain (Fig. 3D). However, across strains producing detectable mCherry, mean expression across the core promoter library displayed a 14-fold expression range, from 0.25 to more than 5-fold pTEF1. These results indicate that, like full-length promoters, core promoter sequences can drive divergent transcriptional outputs in *A. oryzae* (Fig. 3D). Proteomics analysis of mCherry in biomass grown in submerged fermentations confirmed that core promoters could also drive protein expression in mycelia, including at levels that were several-fold higher than pTEF1 (Supplementary Fig. 9A). There was a significant correlation between the flow cytometry and proteomics data ($r = 0.82$, $R^2 = 0.67$, $p < 0.01$, Supplementary Fig. 9B). This suggests that flow cytometry is a useful screening approach to identify constitutive promoters. Nonetheless, following up on flow cytometry screening results in mycelia may be needed for establishing the precise promoter strength for submerged fermentations.

To further benchmark the SES system we used proteomics to compare the strength of the SES promoter Ao_0583, identified as >4-fold stronger than pTEF1 in both conidia and mycelia, with the starch-inducible pAmyB promoter, one of the strongest known endogenous *A. oryzae* promoters that is frequently used for high protein expression and secretion from submerged cultures[21,50]. Strikingly, Ao_0583 was approximately 6-fold stronger than that of the pAmyB promoter when the strain harboring it was grown under inducing conditions. mCherry levels were estimated to comprise approximately 13% intracellular protein under the Ao_0583 system, and only 1–2% in the pAmyB expression strain (Fig. 3E and Supplementary Fig. 10). While mCherry levels did not differ between glucose and dextrin when using the SES, pAmyB expression increased on dextrin, the predicted inducer of pAmyB (Fig. 3E and Supplementary Fig. 10). Thus, the SES permits high protein expression independently of the carbon source and avoids the complex multi-step regulation and global metabolic changes involved in pAmyB-driven expression in *A. oryzae*[60]. To our knowledge, the expression levels afforded by the SES far outperform any characterized promoter in *A. oryzae*.

In addition to the limited set of available mono-directional promoters, there is a lack of bidirectional promoters for filamentous fungi. Bidirectional promoters, which are available in yeast, could accelerate genetic engineering in edible filamentous fungi by enabling assembly of multi-step metabolic pathways or multi-protein complexes through fewer transformations[61,62]. We addressed this challenge in two ways. First, we created a synthetic bidirectional promoter (Syn-BD), as the modular nature of the SES enables different parts to be combined to create highly programmable modes of gene expression[63]. By combining two core promoters (*gpdA* and *hhfA*) with 2× UAS binding sites, we created a 485-bp bidirectional promoter was sufficient to drive bidirectional gene expression using the SES (Fig. 3F). Second, we computationally identified candidate endogenous bidirectional promoters using publicly available RNAseq data collected across diverse growth conditions (Supplementary Data file 2). Out of five computationally identified bidirectional promoter candidates (Supplementary Table 7), two (p2-1 and p4-2) could drive mCherry and GFP expression in a bidirectional fashion (Supplementary Fig. 11). The p4-2 promoter displayed similar levels of expression as the rationally designed Syn-BD and a control bidirectional promoter composed of concatenated pAmyB-pTEF1 sequences pointing in separate directions (Fig. 3G). Interestingly, the sequence of the identified p4-2 endogenous promoter in *A. oryzae* has the same length and genomic context as the H3/H4 histone promoter which was previously identified and evaluated in a range of *Aspergilli*, but not in *A. oryzae*[64]. This suggests that our computational pipeline might be broadly useful to identify bidirectional promoters in filamentous fungi. Overall, these results expand the set of gene regulation tools and promoters available for engineering the edible *A. oryzae*.

## Edible mycelium bioengineered for enhanced nutritional value and sensory appeal

Having established a synthetic biology toolkit for *A. oryzae*, we next sought to deploy our tools to bioengineer its edible mycelium, as a first

step toward enhancing its value as mycoprotein. We were inspired by the recent commercial success of bioengineered *Saccharomyces* used for brewing, which have been modified for improved cost savings, sustainability, and sensory profiles[17]. To explore whether genetic modification could similarly enhance foods made with filamentous fungi, we set out to modify endogenous biosynthetic pathways that could potentially improve the nutritional value and sensory appeal of the edible *A. oryzae* mycelium for alternative meat applications.

We initially focused our engineering efforts on ergothioneine, a bioactive amino acid and powerful antioxidant. Low plasma levels of ergothioneine are correlated with cardiovascular disease and neurological decline, and humans encode a specific ergothioneine transporter that uptakes ergothioneine from the diet, underscoring the potential importance of this molecule in human health[65]. While many foods contain low levels of ergothioneine, fungi are the major dietary source[66]. Work in the model ascomycete mold *Neurospora crassa* has revealed that fungal ergothioneine biosynthesis involves two enzymes, Egt1 and Egt2, that convert cysteine, S-adenosylmethionine, and histidine, to ergothioneine[67–69] (Fig. 4A). *N. crassa* Egt1 and Egt2 were recently co-expressed in *A. oryzae* using plasmid-based random integration[70]. Rice cultured with transformants in solid-state fermentation had elevated ergothioneine levels, but the levels in the biomass alone were not investigated. Untransformed *A. oryzae* produced low levels of ergothioneine, suggesting that this fungus may harbor endogenous pathways for production[70].

Instead of introducing foreign genes, we hypothesized that by changing the expression of potential endogenous *A. oryzae* genes involved in ergothioneine biosynthesis, we could elevate production in

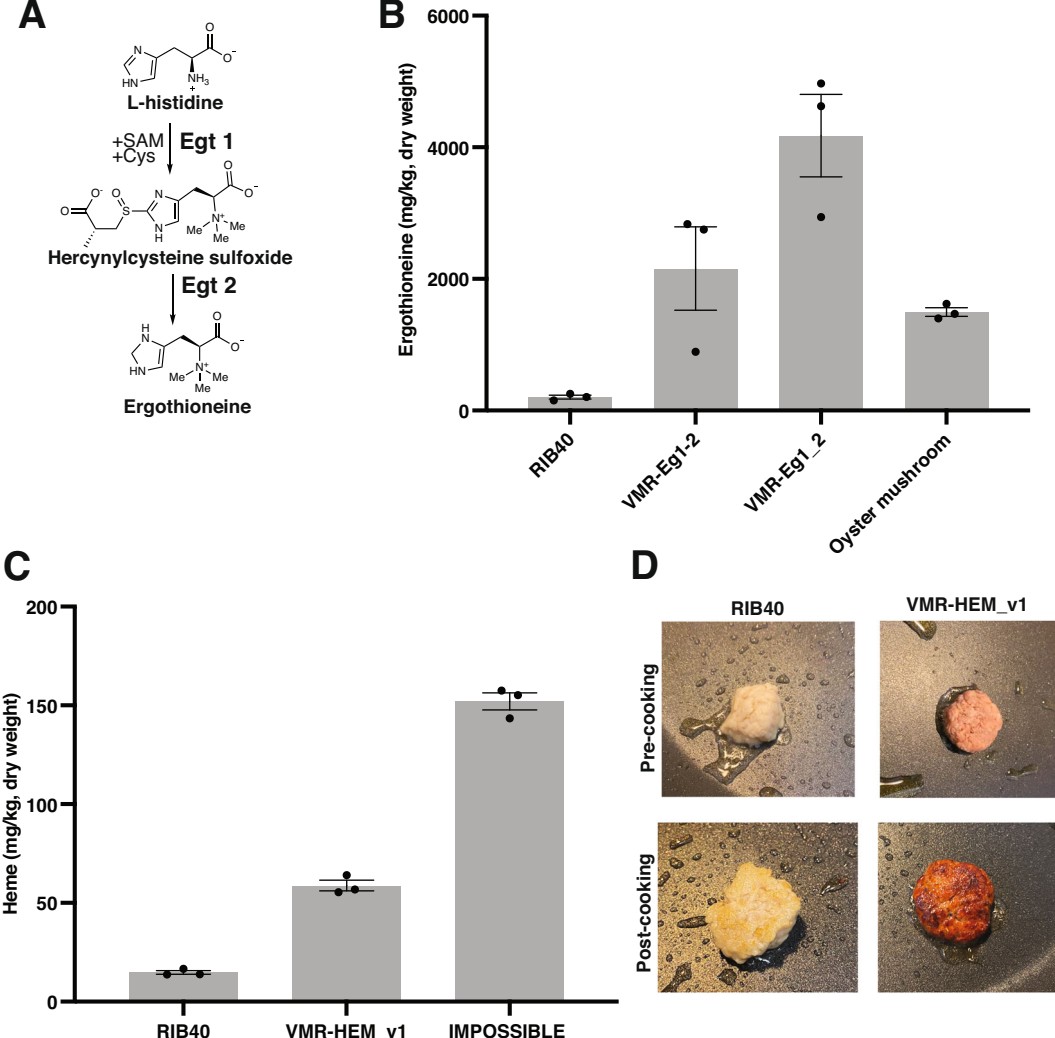

**Fig. 4 | *A. oryzae* mycelium bioengineered for ergothioneine and heme content.** **A** Fungal biosynthesis of ergothioneine, a powerful antioxidant associated with several health benefits in humans. The characterized biosynthetic pathway from the fungus *Neurospora crassa* involves the enzymes Egt1 and Egt2. **B** *A. oryzae* homologs of *N. crassa* Egt1 and Egt2 were identified bioinformatically (see Supplementary Table 7 and Supplementary Fig. 12 for details) and expressed from neutral loci using a bidirectional promoter (strain VMR-Eg1-2) or as two separate genes at two different genomic locations, with each gene under the control of its own promoter (strain VMR-Eg1_2). The strategy is described in Supplementary Fig. 12. Oyster mushroom, the dietary mushroom with the highest ergothioneine content, was included for comparison. Biomass was analyzed by LC–MS. Results are average and SEM of three biological replicates. **C** Engineering of heme biosynthesis in *A. oryzae* biomass. The strategy is described in Supplementary Fig. 15. Heme was quantified using LC–MS in the biomass. The intracellular heme levels in the engineered strain were 4-fold higher than in the background strain, RIB40, and 40% of those found in IMPOSSIBLE™ burger made from plants, a leading plant-based meat product incorporating heme for flavor and color, was included for comparison. Results are average and SEM of three biological replicates. **D** Color of harvested background and engineered heme strain after culturing. The engineered strain overproducing heme (VMR-HEM_v1) was distinctly red in color, while RIB40 was off-white. The harvested fungal biomass could be readily formulated into an imitation meat patty with minimal processing. The color difference remained upon cooking, further enhancing the meat-like appearance of the naturally textured fibrous biomass.

the edible biomass to levels found in dietary mushrooms. To identify candidates, we searched the *A. oryzae* genome for homologs of *N. crassa* Egt1 and Egt2. We found two *A. oryzae* ortholog candidates sharing 49.2 and 45.1% amino acid sequence (Supplementary Table 8). Sequence alignment indicated conservation of key residues or functional groups bioinformatically predicted to be involved in substrate binding in Egt1, as well as residues structurally confirmed to participate in catalysis in Egt2[68] (Supplementary Fig. 12). We then integrated the *A. oryzae* homologs (named AO_Egt1 and AO_Egt2) at neutral loci and drove expression of both genes from either a bidirectional promoter (strain VMR-Eg1-2), or as two separate genes in two separate genomic locations (strain VMR-Eg1_2) (Supplementary Fig. 12). High-resolution Liquid Chromatography-Mass Spectrometry (LC−MS) was used to detect ergothioneine in samples (Supplementary Fig. 13). Consistent with previous observations[70], we detected low levels of ergothioneine in the mycelium in the background strain RIB40 (Fig. 4B and Supplementary Fig. 13). However, the bidirectional promoter and separate promoter strains elevated ergothioneine 11-fold and 21-fold, respectively, over RIB40 (Fig. 4B). While the ergothioneine levels in VMR-Eg1-2 was similar to those found in oyster mushroom, the highest known dietary ergothioneine source[66], the mean levels in strain VMR-Eg1_2 were 1.5-fold higher. We observed no major difference in protein content between the wild-type and engineered strains; however, ergothioneine overproduction was associated with a slight growth defect, suggesting a metabolic burden of ergothioneine production under the growth conditions (Supplementary Fig. 14). Overall, these results implicate the endogenous genes AO_Egt1 and AO_Egt2 in ergothioneine biosynthesis in *A. oryzae* and validate the metabolic engineering approach to alter the molecular composition of mycoprotein.

Having validated our tools to increase levels of bioactive molecules for enhanced nutritional value, we asked whether a similar approach could be applied to sensory properties of the edible biomass to more closely mimic animal meat. For example, even though the *A. oryzae* biomass has a meat-like fibrous texture owing to its microscopic morphology, the biomass, which is off-white, would necessitate color addition for many meat applications. As a first step toward improving the meat-like flavor composition and appearance of the edible biomass using bioengineering, we initially targeted the biosynthesis of heme, an essential cofactor that catalyzes a wide range of reactions across all domains of life and gives red meat its color and contributes to flavor upon cooking[71]. IMPOSSIBLE Foods, a leading plant-based meat producer, has taken advantage of these properties of heme and adds a purified soy Leghemoglobin (LegH) produced with the yeast *Pichia pastoris* to its products based on plant protein isolates to create realistic alternatives that look like red meat[72,73]. Other plant-based meat producers have now followed suit with similar hemoglobin addition strategies[74].

We reasoned that by modulating the expression of key heme biosynthetic enzymes, we could elevate intracellular heme in the edible fungal biomass to levels found in leading meat alternatives incorporating heme for flavor and color. Fungal heme biosynthesis is carefully regulated at the transcriptional and post-translational levels and involves eight dedicated enzymes, which are split between the mitochondria and the cytosol[75] (Supplementary Fig. 15). We identified potential heme biosynthesis proteins in *A. oryzae* by searching the genome for sequences found in *S. cerevisiae*[76] (Supplementary Table 9). There is limited experimental information about heme biosynthesis in filamentous fungi, but based on successful engineering efforts from yeast[77,78], and studies of individual heme biosynthetic enzymes in different *Aspergilli*[75,79–81], we initially targeted expression of predicted rate-limiting enzymes, including ALAS (biosynthetic enzyme#1), PBGD (#3), UROD (#4), and CPO (#5). Additionally, we mutated key cysteine residues in the Heme Regulatory Motif (HRM) of ALAS to remove potential feedback inhibition by heme[82]

(Supplementary Fig. 15). Importantly, high levels of free heme and the porphyrin intermediates can be toxic to the cell, causing oxidative damage and hampering growth[77,83]. To address this potential challenge, we expressed two copies of Soy Leghemoglobin, the FDA-approved protein used in IMPOSSIBLE meat[72], as a potential heme sink, using both the SES and the significantly weaker pTEF1 promoter. Though the regulation of heme biosynthesis has not been characterized in detail in filamentous fungi, simultaneous elevation of biosynthetic enzymes and a heme-binding protein was necessary to increase heme levels without causing excessive toxicity in *S. cerevisiae*[77,78]. The final engineered strain *A. oryzae* contained a total of separate six modifications (Supplementary Fig. 15).

We used high-resolution LC−MS to detect heme across all samples (Supplementary Fig. 16). The biomass of the engineered strain contained 4-fold higher levels of heme compared to the non-engineered strain, on a dry weight basis. These levels of heme in the engineered strain were nearly half (40%) of those found in IMPOSSIBLE meat (Fig. 4C). Increasing levels further may require tuning pathway flux or making additional modifications beyond biosynthetic enzyme expression levels, as was recently shown in *S. cerevisae*[76]. However, to our knowledge, this is the highest levels of intracellular heme in fungal mycelium and a rare example of heme biosynthesis engineering in filamentous fungi. Given the importance of heme for enzyme production for biofuels and medical applications[78,84], we envision that these strains and approaches could have broad applicability for engineering efforts beyond food.

Upon harvesting the biomass of the heme overproducer, we noticed that it was red in color, compared to the off-white color of the background strain. In contrast to other plant-based meat alternatives, which require extensive processing and ingredient addition to transform off-flavor plant protein isolates (such as soy or pea) to meat alternatives, this bioengineered mycoprotein required minimal post-harvest processing for formulation into an imitation red meat patty following a standard mycoprotein production protocol[11] (Fig. 4D). The only processing needed was removing excess liquid from the biomass prior to grinding and cooking. The color difference between the background and engineered strains remained after cooking, enhancing the meat-like appearance of the naturally textured, fibrous fungal biomass (Fig. 4D). There was no decrease in the growth yield or protein content (46%, on a dry weight basis) in the engineered heme strain relative to the background strain (Supplementary Fig. 17). The engineered mycoprotein also contained all the essential amino acids, suggesting a promising nutritional profile (Supplementary Fig. 17). Taken together, these data suggest that the engineered edible fungal mycelium could have promise in meat alternative applications.

## Discussion

Filamentous fungi are widely used for the industrial production of enzymes and metabolites and recently have found more widespread use in both sustainable materials and foods[12]. However, genetic tools for these organisms have historically been limited in both sophistication and scope, preventing both engineering efforts and fundamental studies. Recent advances in CRISPR-Cas9 technology have dramatically improved the possibilities of modifying diverse mushrooms and molds, including the food-safe, edible fungus *A. oryzae*[27–29,34]. In contrast to industrial strains such as *T. reesei*, which was recently used to produce milk and egg proteins at lab scale[14], the historically consumed *A. oryzae* has potential uses across fermented foods, food protein production, cellular agriculture, and mycoprotein[13,20,22,23].

To enable bioengineering for these diverse applications, we developed a ready-to-use toolkit that is now available to the research community and includes DNA parts for integration and regulation of genes and pathways. Similar toolkits are available in *S. cerevisiae*, where they have significantly expanded opportunities for genetic engineering[45]. We hope that our tools will be similarly useful for

expanding the engineering possibilities in *A. oryzae*, alongside other recently developed genome modification methods such as base editing[85], in vivo DNA assembly in NHEJ-deficient strains[86], and protein expression screening[87]. Additionally, we envision that the computational and experimental approaches used here – for identification and evaluation of neutral loci and design and identification of promoters – could be broadly useful for constructing genetic toolkits and engineering diverse fungal hosts.

We used our tools to enhance the molecular composition and appearance of the mycelium as a first step toward improving its nutritional and sensory properties. First, we engineered *A. oryzae* mycoprotein to overproduce the nutraceutical ergothioneine at levels that are higher than those in mushrooms, the highest known natural source from the diet. While ergothioneine has been produced in a range of microbial hosts for the purpose of isolating the nutraceutical[88,89], our work represents a proof of concept of modifying endogenous ergothioneine biosynthesis for mycoprotein applications. Separately, we engineered the *A. oryzae* mycelium to overproduce heme, a key flavor-and-color molecule in red meat, at levels that are close to half those found in leading plant-based meats. Our engineering of the edible fungal biomass for alternative meat presents an alternative approach to fungal food beyond the production of secreted animal proteins, which is a less efficient fermentation process and has a higher environmental footprint than biomass production[6,9]. Future engineering targets for edible fungal biomass could include lipid pathways for flavor, amino acids for nutrition, structural alteration for texture improvement, or enzymes for improved growth on affordable, complex feedstocks. However, it is important to note that our work represents early prototypes, and further assessment of the sensory attributes, consumer acceptability, potential food safety concerns, and the regulatory landscape around genetically modified organisms (GMO), is needed to bring engineered edible fungi from lab bench to the table.

*A. oryzae*, like many of the strains that form the basis of fungal foods available on the market, has been genetically modified through extensive selection and breeding throughout human history[90,91]. Genetic modification using contemporary gene editing tools such as CRISPR-Cas9 represents a natural next step in this long history of microbial gene modification to suit human needs and holds promise to further expand fungal strain diversity and accelerate the adaptation of fungal strains to the demands of current production methods and consumer preferences. Bioengineered edible plants and yeasts have demonstrated reduced environmental impact, improved nutrition, and improved flavor profiles compared to their non-engineered counterparts and are already available on the market[17,92,93]. We anticipate similar possibilities with genetic modification of edible filamentous fungi, as synthetic biology in these organisms is uniquely positioned to address the pressing environmental, ethical, and public health challenges of industrial animal agriculture.

## Methods
### Cloning
All primers used for genome modification are shown in Supplementary Table 10. All strains and plasmids used for strain construction are listed and described in Supplementary Tables 11-12. The sequence files corresponding to each strain and plasmid can be found in the JBEI Public Registry (https://public-registry.jbei.org/)[94]. All plasmids were propagated in *Escherichia coli* strain DH10B and purified by Miniprep (Qiagen). The plasmids generated in this study were based on the pTWIST_amp backbone (TWIST biosciences) and were constructed by Gibson assembly[95] using Gibson assembly master mix (New England Biolabs). PCR amplification was performed using NEB Q5 polymerase according to the manufacturer's instructions (New England Biolabs). All genes were codon optimized for *A. oryzae* and ordered either as G-blocks from IDT or as complete, sequence-verified genes from IDT

or TWIST biosciences. The coding sequences of heterologous genes in all plasmids were validated by Sanger sequencing (Azenta) or whole-plasmid sequencing (Primordium).

### Growth conditions
*A. oryzae* strains were always grown at 30 °C. A variety of media were used in the transformation and cultivation of *A. oryzae*, and these are indicated below. They are referenced in the materials and methods section. Media were supplemented with 5 g/L uridine (Sigma–Aldrich, #U6381) or 2 g/L uracil (Sigma–Aldrich, #U1128) when supplementation to support growth of *pyrG* mutants was needed. Supplementation is indicated as UU throughout the manuscript.

**GP medium (per 1 L of medium).** 5 g yeast extract, 10 g polypeptone, 0.5 g $MgSO_4 \cdot 7H_2O$, 5 g $KH_2PO_4$. 20 g glucose was used as the carbon source unless otherwise indicated. Alternatively, 20 g dextrin (Sigma–Aldrich, #31400) was used as the carbon source.

**PDA + 5-FOA + Uridine + Uracil (PDA + 5-FOA + UU) medium (per 1 L of medium).** 39 g Potato Dextrose Agar (PDA, Sigma–Aldrich, #70139-500 G), 5 g uridine, 2 g uracil, and 1 mg/mL 5-fluoroorotic acid (ThermoFisher, #R0812).

**Bottom Agar + Methionine (BA + Met, per 0.5 L of medium).** 1 g $NH_4Cl$, 0.5 g $(NH_4)_2SO_4$, 0.25 g KCl, 0.25 g NaCl, 0.5 g $KH_2PO_4$, 0.25 g $MgSO_4 \cdot 7H_2O$, 0.01 g $FeSO_4$, 109.3 g sorbitol, 7.5 g agar, 10 g glucose, 0.75 g methionine. pH was adjusted to 5.5 prior to autoclaving.

**Top Agar + Methionine (TA + Met, per 0.5 L of medium).** Same as BA + Met, but 4 g agar instead of 7.5 g agar per 0.5 L.

**Minimal Medium Agar + Methionine (MMA + Met, per 0.5 L of medium).** Same as BA + Met, but no sorbitol added as the osmotic stabilizer.

**CDA medium (per 1 L of medium).** 3 g $NaNO_3$, 2 g KCl, 1 g $KH_2PO_4$, 0.5 g $MgSO_4 \times 7\, H_2O$, 0.02 g $FeSO_4 \cdot 7H_2O$, 15 g agar. 20 g glucose was used as the carbon source unless otherwise indicated. Alternatively, 20 g dextrin (Sigma–Aldrich, #31400) was used as the carbon source.

**CDA(Leu) medium.** same as CDA medium but containing 10 mM leucine as the sole nitrogen source instead of the 3 g/L $NaNO_3$.

### Strain construction
*A. oryzae* strains were genetically modified using protoplast transformation (see standard transformation protocol below). All *A. oryzae* strains are described in Supplementary Table 12. Regenerated protoplasts were restreaked onto MMA + Met plates to obtain single colonies and purify the potentially heterokaryotic conidia. Following 48 h of growth at 30 °C, the conidia of individual, single colonies were transferred to MMA + Met slants for growth for 48–72 h at 30 °C. These purified strains represented the strains used in all assays and characterizations. To confirm the insertion of genes at the correct locus, colony PCR was performed on conidia on slants using PHIRE direct plant PCR kit (ThermoFisher, #F130WH) by boiling conidia in 20 μL of dilution buffer for 10 min at 95 °C and using 1 μL of the conidial spore suspension as the template for PCR, which was set up according to the manufacturer's instructions. Strains harboring the correct insertions were saved as glycerol stocks by suspending conidia in 30% glycerol (v/v). For the simultaneous targeting of *wA* and *niaD* loci in a single transformation, DNA templates of plasmids were prepared as described below and 10 μg of each plasmid, along 5 μL of each RNP complex (four total, two per locus) were added at the DNA-RNP incubation step. To check for spore coloration (*wA* and *yA* mutants), strains were grown on PDA medium for 5 days at 30 °C. To check for nitrate assimilation (*niaD*

mutant), strains were grown on CDA and CDA-Leu for 5 days at 30 °C. To check for *pyrG* mutation and the associated uridine/uracil auxotrophy, strains were grown on CDA and CDA supplemented with 2 g/L uracil and 5 g/L uridine for 5 days at 30 °C. To assess the targeting efficiency at individual loci, and to evaluate the effect of homology arm length on integration at the *wA* locus, colony PCR of 10 individual strains was performed, and those displaying the correct band by PCR were deemed successful integrations. Varying homology arm lengths of the *wA* fixing template were obtained by linearizing the full-length template with different primers (see primer table, Supplementary Table 10). Flow cytometry or microscopy assays were used to assess the expression of fluorescent proteins (see below for details). To create the engineered strain VMR-Eg1-2, *A. oryzae* RIB40 *pyrG* mutant was transformed with the linear DNA template originating from JBx_250940 and the two 5′ and 3′ RNP complexes targeting the *chro1-3* neutral locus. To create the engineered strain VMR-Eg1_2, *A. oryzae* RIB40 *pyrG* mutant was transformed with the linear DNA template originating from JBx_250936 and the two 5′ and 3′ RNP complexes targeting the chro1-3 neutral locus, and subsequently with the linear DNA template originating from JBx_250938 and the two 5′ and 3′ RNP complexes targeting the *chro2-2* neutral locus. For the engineered strain overproducing heme (VMR-HEM_v1), *A. oryzae* RIB40 *pyrG* was sequentially transformed with linearized DNA originating from plasmids JBx_250942, JBx_250944, JBx_250946, JBx_250948, JBx_250950, JBx_236225, as well as the specific 5′ and 3′ RNP complexes associated with the target locus for integration. For the cultivation of RIB40, and the engineered strains overproducing ergothioneine and heme, $5 \times 10^5$ conidia were inoculated into 50 mL of GP-glucose medium (ergothioneine strains and corresponding RIB40 control) or 50 mL GP-dextrin medium (heme strain and corresponding RIB40 control) in 250 mL Erlenmeyer flasks. The strains were grown for 96 h at 30 °C, shaking at 160 rpm. Biomass was harvested by vacuum filtration over Miracloth. Biomass was lyophilized for extraction of metabolites and was dried for 7 days at 50 °C prior to recording of the dry mass.

### Transformation of *A. oryzae*
**Preparation of linearized DNA fixing templates for transformation.** To generate linear DNA to be transformed into *A. oryzae* as fixing templates alongside CRISPR-Cas9 RNP complexes, the DNA was linearized using PCR from the corresponding plasmids harboring the DNA fixing template of interest. Briefly, 1 ng of plasmid DNA was used as the template for a 60 μL PCR reaction using the Q5 high-fidelity polymerase master mix (New England Biolabs, #M0492S) and following the manufacturer's instructions for the PCR protocol. For each DNA template, five 60 μL reactions were set up in parallel to obtain sufficient DNA for transformation. The PCR reactions were then combined and purified using the QIAquick PCR purification kit (Qiagen) and were eluted at the final step in 25 μL of sterile water. This typically gave sufficient quantities of the large amount of DNA needed for protoplast transformation (>10 μg in 20 μL).

**Preparation of RNP complexes for transformation.** All CRISPR-Cas9 reagents were obtained from IDT, including the Alt-R S.p. HiFi Cas9 Nuclease V3 (IDT, #1081061), Alt-R CRISPR-Cas9 crRNA XT 2 nmol (customized sequence), and Alt-R CRISPR-Cas9 tracrRNA 100 nmol (IDT, #1072534). RNA duplex buffer was included as part of the tracrRNA. The crRNA, which is the sequence-specific RNA that targets the region of interest, was resuspended in 20 μL water for 100 μM final concentration. To hybridize the sequence-specific crRNA to the universal tracrRNA to generate the final sgRNA, 0.5 μL of crRNA (5 μM final concentration) was mixed with 0.5 μL tracrRNA (5 μM final concentration) in 9 μL RNA duplex buffer and the mixture was then heated for 5 min at 95 °C and was then left to cool. This hybridized mixture represents the final sgRNA which is ready to bind to the Cas9 protein to form the RNP complex. To create the final RNP complexes for transformation, 2.16 μL of the hybridized crRNA-tracrRNA (the sgRNA, final

concentration 540 nM) was mixed with 0.18 μL of Alt-R S.p. HiFi Cas9 Nuclease V3 (final concentration 540 nM) and 17.66 μL buffer (9 mM HEPES, 67 mM KCl, pH 7.5). The mixture was incubated at room temperature for 20 min to form the RNP complex and was then transferred to ice. As specified below in the transformation protocol, 5 μL of each RNP complex was used per 200 μL protoplast transformation. The RNP complexes were always prepared fresh on the day of the transformation and were never subjected to freeze-thaw.

**Standard transformation protocol.** Protoplast-mediated transformation was used to transform *Aspergillus oryzae*. We followed the protocol from[28], with minor modifications. To generate mycelial biomass for protoplast generation, *pyrG* mutant strains were grown in duplicate in 50 mL GP medium supplemented with uracil and uridine in 250 mL flasks at 30 °C, shaking at 160 rpm. Following 72 h of growth, mycelia were harvested by pressing the liquid from the mycelia in a 20 mL syringe harboring a sterile cotton ball.

Dry mycelia from one flask were then put in 10 mL of TF1 solution (Per 500 mL of water: 2.9 g maleic acid, 39.5 g $(NH_4)_2SO_4$, pH adjusted to 5.5 and filter sterilized) harboring 0.1 grams of YATALASE enzyme (Takara Bio, #T017) which was used to digest the cell wall. This was incubated shaking for two and a half hours at 30 °C and 160 RPM, and the tissue was pressed within a sterile syringe with cotton to collect the protoplasts within the flow-through. The flow-through was checked for cloudiness, which indicated the generation of protoplasts. The resulting protoplast solution (7–8 mL usually, as sometimes not all protoplast solution could be pushed through the cotton ball due to clogging) was centrifuged at 475 g for 10 min, and the supernatant was then discarded. The protoplasts were then gently resuspended in 10 mL of TF2 solution pre-warmed at 30 °C (Per 1 L of water: 218.5 g sorbitol, 10.95 g CaCl$_2$•6H$_2$O, 2.05 g NaCl, 1.21 g Tris buffer, pH adjusted to 7.5 and filter sterilized). The protoplast solution resuspended in TF2 was centrifuged at 475 g for 10 min to discard the supernatant. The protoplasts were resuspended in 1–2 mL of TF2 solution and 200 μL of protoplast solution was placed into 15 mL volume centrifuge tubes. At least $1.2 \times 10^7$ protoplasts/mL was needed, so it is useful to consider this concentration when resuspending in the 1–2 mL TF2 solution. Concentrations of ~$10^8$ protoplasts per mL were typically obtained in the standard digestion protocol.

To transform the protoplasts, a total of 10 μg of PCR-linearized DNA fixing template (in 20 μL sterile water) was added to the 200 μL protoplast solution. Then 5 μL of each of the two pre-formed sgRNA-Cas9 RNP complexes were added and the protoplast-DNA-RNP mixture was incubated on ice for 30 min. After 30 min, sequentially and slowly, 250 μL, 250 μL, and 850 μL aliquots of TF3 solution were added (TF3 solution: per 1 L of water: 600 g Polyethylene glycol (4000), 10.95 g CaCl$_2$•6H$_2$O, 1.21 g Tris buffer, pH adjusted to 7.5 followed by autoclaving) and left to incubate at room temperature for 30 min. A total of 5 mL of TF2 solution was then added to each protoplast solution then centrifuged for 10 min at 475 g to discard the supernatant. The protoplast pellet was resuspended in 500 μL of TF2. At this point, a bottom agar plate pre-warmed at 30 °C already brought out from its heating location (see above for bottom agar recipe). The 500 μL of protoplast suspension was then mixed with 5 mL of a liquid layer of top agar (see above for recipe) pre-warmed to 50 °C, then quickly spread uniformly across the bottom agar and left to solidify at room temperature. Transformants were left for 72 h to regenerate the protoplasts. Transformants were then restreaked according to the procedure described in "strain construction" above.

**Miniaturized transformation protocol.** To speed up and miniaturize the transformation protocol to make it compatible with a 96-well plate format, it was modified according to the process below. Protoplasts were generated by digestion according to the standard protocol. However, only 50 μL protoplasts were used for the transformation, and

1.75 µL of each RNP complex was added alongside 2.5 µg DNA to these protoplasts and the mixture was incubated on ice for 30 min as in the standard protocol. Then, only 212.5 µL TF3 solution was added to the DNA-RNP-protoplast solution, and the entire mixture was spread on a bottom agar plate following the standard washing, using an L-shaped spreader. No top agar was used in the regeneration of the protoplasts. All steps following the plating of protoplasts were the same as described above in the standard protocol. All results reported in the paper followed the standard transformation protocol unless otherwise indicated.

**Generation of pyrG mutants.** The two transformation protocols above describe how to transform *pyrG* strains using a linear DNA fixing template. To generate *pyrG* mutants in the first place, as we demonstrated with five different *A. oryzae* strains obtained from NRRL, the standard protocol was changed slightly according to the following modifications. No linear DNA was transformed. Only the two RNP complexes targeting the *pyrG* gene were added to incubate on ice with the protoplasts. At the final step, strains were plated onto top and bottom agar supplemented with 1 mg/mL 5-FOA 2 g/L uracil, and 5 g/L uridine. Following the regeneration of protoplasts, instead of being restreaked on MMA + Met plates for single colonies and then transferred to MMA + Met slants, strains were restreaked on MMA + Met supplemented with 1 mg/mL 5-FOA and 2 g/L uracil and 5 g/L uridine. Surviving colonies were transferred to slants with the same medium. Between two and three colonies of each strain were analyzed for auxotrophy by plating on CDA, CDA + Uracil + Uridine. One of these strains was subjected to further analysis by amplification of the *pyrG* gene and flanking regions using primers pyrG-2F and pyrG-2R. The 2640-bp amplicon was purified and subjected to sequencing. The sequences were aligned using Snapgene.

**Excision of pyrG marker.** To set up the excision to remove the *pyrG* marker via locus-specific recycling, we followed the protocol described in ref. 96 with minor modifications: spore suspensions were generated by adding 0.5–1 mL sterile water to MMA + Met slants harboring single colonies subjected to colony PCR. Slants were vortexed to resuspend conidia, and then 500 µL of suspension harboring between $10^5$ and $10^6$ conidia/mL was spread onto PDA + 5-FOA + Uridine + Uracil plates. The conidia were spread using an L-shaped spreader and plates were left to dry for 1–2 h. The plates were then incubated at 30 °C for 5–7 days, at which point healthy, robustly growing colonies appeared on the plates. Conidia from individual colonies that appeared healthy were transferred to PDA + 5-FOA + uridine + uracil slants, whereby they were subject to an additional 4–5 days of growth. These strains had the *pyrG* marker excised. To verify the marker excision, conidia from slants were subjected to colony PCR (as described in standard transformation protocol), or PCR on extracted genomic DNA. To extract genomic DNA, a small amount of conidia was transferred to 300 µL lysis buffer (2% Triton X-100, 1% SDS, 100 mM NaCl, 1 mM EDTA, 10 mM Tris pH 8) in a bead-beating tube (Lysing Matrix Z, MP Biomedicals, catalog#: 116961050-CF). Bead beating was performed for 1 min. Then, samples were incubated at 65 °C for 30 min, vortexing every 10 min. 300 µL of phenol:chloroform:isoamyl alcohol 25:24:1 reagent was then added (Sigma–Aldrich, #P3803) and tubes were vortexed for 5 min and were then centrifuged at max speed for 10 min to separate the layers. 120 µL of the top aqueous layer was transferred to a new tube, and 210 µL of ice-cold 100% ethanol was added to precipitate the DNA. The DNA pellet was washed twice with 70% and was resuspended in 30 µL sterile water. 1 µL was used as the template for PCR reactions using the PHIRE direct plant PCR kit (ThermoFisher, #F130WH).

**Whole-genome sequencing, assembly, annotation, and phylogenetic analysis of diverse *A. oryzae* strains obtained from NRRL Sequencing.** *A. oryzae* strains subjected to sequencing were obtained from NRRL (NRRL numbers: #2215, #5592, #32614, #1911, #6574). They were grown in GP-glucose medium (50 mL medium in 250 mL flasks) for 72 h prior to harvesting by vacuum filtration and flash-freezing in liquid nitrogen. gDNA extraction, sample quality assessment, DNA library preparation, sequencing, and bioinformatics analysis were conducted at Azenta Life Sciences. Genomic DNA was extracted using DNeasy Plant Mini Kit following manufacturer's instructions (Qiagen). Genomic DNA was quantified using the Qubit 2.0 Fluorometer (ThermoFisher Scientific). NEBNext® Ultra™ II DNA Library Prep Kit for Illumina, clustering, and sequencing reagents was used throughout the process following the manufacturer's recommendations. Briefly, the genomic DNA was fragmented by acoustic shearing with a Covaris S220 instrument. Fragmented DNA was cleaned up and end repaired. Adapters were ligated after adenylation of the 3'ends followed by enrichment by limited cycle PCR. DNA libraries were validated using a High Sensitivity D1000 ScreenTape on the Agilent TapeStation (Agilent Technologies) and were quantified using Qubit 2.0 Fluorometer. The DNA libraries were also quantified by real-time PCR (Applied Biosystems). The sequencing library was clustered onto lanes of an Illumina HiSeq 4000 (or equivalent) flow cell. After clustering, the flow cell was loaded onto the Illumina HiSeq instrument according to the manufacturer's instructions. The samples were sequenced using a 2 × 150 bp Paired End (PE) configuration. Image analysis and base calling were conducted by the HiSeq Control Software (HCS). Raw sequence data (.bcl files) generated from Illumina HiSeq was converted into FastQ files and de-multiplexed using Illumina bcl2FastQ 2.17 software. One mismatch was allowed for index sequence identification.

**Assembly, annotation, and prediction of biosynthetic gene clusters.** The reads were filtered with TrimmomaticPE version 0.39[97] with the following parameters: LEADING:30 TRAILING:30 MIN-LEN:120. The filtered reads were used for de novo assembly using the SPAdes[98] genome assembler v3.13.1-1 with the following parameters --careful --cov-cutoff 100. The resulting assemblies were then processed with AUGUSTUS[99] v3.4.0, to obtain coding sequences and protein predictions. Augustus was executed using a gene model for *Aspergillus oryzae* to identify start and stop codons, introns, and exons. For the prediction of natural product production repertoire of the strains, the assembled genomes and their gene calling files were used for functional annotation and mining for natural products biosynthetic gene clusters using antiSMASH version 7[100].

**Phylogenetic analysis.** The taxonomic affiliation of the *A. oryzae* strains used in this study (tree in Supplementary Fig. 2) was defined using a multilocus phylogenetic tree constructed with the genomes of 59 Aspergillus spp. strains which were obtained from the GenBank database. These genomes were processed with AUGUSTUS[99] v3.4.0, to obtain coding sequences and protein predictions. The predicted proteomes of the Aspergillus dataset. Given the closeness of the strains, a genome from a distantly related taxonomic group (*Trichoderma atroviridae*) was added to the dataset to reduce the number of shared orthologs. The core genome was then calculated using BPGA[101] this analysis led to a set 237 conserved proteins that were sorted, aligned[102], and trimmed[103], after this process 167 protein sequences remained. They were then concatenated, and an evolutionary model was calculated for each of the 167 protein partitions. Then a phylogenetic tree was calculated with IQtree2 v2.0.7[104] using maximum likelihood with 10,000 bootstrap replicates. The entire process was executed automatically using a script available at https://github.com/WeMakeMolecules/Core-to-Tree.

## Computational identification of candidate neutral, highly transcribed integration sites for protein expression

These sites were identified from a dataset deposited under BioProject accession: "PRJDB8293". This set of Illumina RNAseq data included 18 libraries which were obtained from *A. oryzae* RIB40 growing in 50 mL cultures in Czapek–Dox liquid medium supplemented with 1% (w/v) Triton X-100 at 30 °C. This dataset was deposited previously by Wong et al.[105]. The specific datasets that were used are shown in Supplementary Table 13. The reads were downloaded from the GenBank FTP using fastq-dump v2.113, the reads were then aligned to the *A. oryzae* reference genome (NCBI RefSeq assembly GCF_000184455.2) using subread package v 2.0.3[106]. To select highly expressed genes, we counted the number of reads that were mapped to each gene in the *A. oryzae* RIB40 genome using featureCounts v2.0.3[107]. The read counts were calculated independently for each run. As the read counts depend on the depth and processing of each sample, a single cutoff cannot be established. Instead, we used this value to rank genes from most expressed to not expressed using the numbers of reads mapped per library (Supplementary Table 13). Then we reasoned that gene that ranked top in all libraries, could be safely considered highly expressed. For selection of *neutral, highly transcribed* integration sites, we first identified all the intergenic regions in the genome and calculated the average read count of the genes flaking them. Then, we sorted these regions from highest to lowest by their average flanking gene read count. For each of the 18 libraries we selected the top 500 intergenic regions and filtered out those that were not found in all conditions; therefore, we selected intergenic regions that are flanked by constitutive, highly expressed genes. This led to a set of 334 regions that were filtered by length (>4.8 kb). Finally, we selected 10 promising intergenic regions spread across *A. oryza*e (Fig. 2, Supplementary Table 4).

## Computational identification of candidate endogenous bidirectional promoters

For the development of new bidirectional promoters for *A. oryzae*, we used the annotated genome of strain RIB40 to mine for all the coding sequences whose start codons are in opposite directions. Then we selected the gene pairs that were highly expressed in most conditions using the Wong et al dataset[105] described above. We then identified the promoter region. We used the same dataset or read counts used for the identification of neutral, highly transcribed integration sites.

## Identification of core promoters for expression in *A. oryzae* using the synthetic expression system

To identify candidate core promoters from the *A. oryzae* genome, we searched for highly expressed genes from publicly available transcriptome data[108]. The gene list of *A. oryzae* RIB40 grown in CD-glucose in liquid cultures without ER stress was sorted by RPKM to generate a list of the top most highly expressed genes. The top eight most highly expressed genes (unique genes, no duplicates) were selected as candidate strong promoters. Additionally, *pdcA*, which appeared at #15 in the rank of this list was selected as other studies suggest this is one of the most highly expressed genes in *A. oryzae*[109]. *thiA*, which was ranked #20 in this list, was also selected, because it has successfully been used for overexpression in *A. oryzae* previously[49]. Finally, *hhfA*, which ranked #55 in the list, was selected, as it was part of the p4-2 bidirectional promoter used in this study. For each of these genes, the genetic DNA 200 bp upstream of the start codon was used as core promoter sequence and ordered as synthetic dsDNA for downstream cloning and transformation. Two additional core promoters that did not come from the transcriptome rank analysis and were not native sequences to *A. oryzae* were also included. These were An_201205 from *A. niger*, which was used previously in the development of a Synthetic expression system in *A. niger* and *T. reesei*[18], as well as the core promoter for Afl_ecm33 from gene AFL2G_04718 in *Aspergillus flavus*. Afl_ecm33 has

been successfully used before to express a secondary metabolite in *A. oryzae* (promoter P4 in this study[48]). For the An_201205 core promoter, the sequence was identical to the one used previously[18], but for Afl_ecm33, the 200 bp upstream of the start codon were selected as the core promoter.

## Flow cytometry assays of conidia for fluorescence quantification

The overall approach followed the method for promoter evaluation described in ref. 47 but with minor modifications. For all flow cytometry assays, strains were grown on PDA + 5-FOA + UU slants (excised neutral loci strains) or PDA slants (all others) for 5–6 days at 30 °C to allow robust development of conidia. Conidia were harvested by the addition of 1 mL of sterile water, followed by vortexing. 250 μL conidial suspension was then transferred to a 96-well plate (Corning, Falcon Tissue Culture Plate, #353072). Flow cytometry assays were performed on the BD Accuri C6 instrument (BD Biosciences) using the following settings: Run limit = 50,000 events, FSC-H threshold <80,000, agitation = 1 cycle every 1 wells. Raw fluorescence data were converted into MEFL (mean equivalents of Fluorescein, for GFP) or METR (mean equivalents of Texas Red, for mCherry), using a fluorescence beads standard (Spherotech, #RCP-30-5A). At least three biological replicates were run for each sample. FlowJo software (version 10) was used to analyze the data.

## Fluorescence microscopy

Strains were grown on either CDA, CDA-Leu, or CDA-dextrin for 4–5 days at 30 °C. Fluorescent protein expression was then imaged in mycelia (edge of the colony) using Leica Microscope DM6B (Leica) and the associated Leica LAS X software (v.5.1.0).

## Proteomic comparison of mCherry expression across media and promoters

**Growth conditions.** to compare the expression of mCherry under endogenous promoters and the core promoters in the synthetic expression system, conidia from three different transformants per construct were inoculated at $5 \times 10^5$ conidia in 50 mL of either CD-dextrin or CD-glucose medium in 250 mL Erlenmeyer flasks. Strains were grown for 96 h at 30 °C, 160 rpm shaking. Biomass was harvested by vacuum filtration over Miracloth and was then lyophilized prior to proteomics.

**Proteomics analysis.** Protein was extracted and tryptic peptides were prepared by following established proteomic sample preparation protocol[110]. Briefly, cell pellets were resuspended in Qiagen P2 Lysis Buffer (Qiagen, Hilden, Germany, Cat.#19052) to promote cell lysis. Proteins were precipitated with addition of 1 mM NaCl and 4× vol acetone, followed by two additional wash with 80% acetone in water. The recovered protein pellet was homogenized by pipetting mixing with 100 mM Ammonium bicarbonate in 20% Methanol. Protein concentration was determined by the DC protein assay (BioRad, Hercules, CA). Protein reduction was accomplished using 5 mM tris 2-(carboxyethyl)phosphine (TCEP) for 30 min at room temperature, and alkylation was performed with 10 mM iodoacetamide (IAM; final concentration) for 30 min at room temperature in the dark. Overnight digestion with trypsin was accomplished with a 1:50 trypsin:total protein ratio. The resulting peptide samples were analyzed on an Agilent 1290 UHPLC system coupled to a Thermo Scientific Orbitrap Exploris 480 mass spectrometer for discovery proteomics[111]. Briefly, peptide samples were loaded onto an Ascentis® ES-C18 Column (Sigma–Aldrich, St. Louis, MO) and separated with a 10 min LC gradient from 98% solvent A (0.1% FA in H2O) and 2% solvent B (0.1% FA in ACN) to 65% solvent A and 35% solvent B. Eluting peptides were introduced to the mass spectrometer operating in positive-ion mode and were measured in data-independent acquisition (DIA) mode with a duty cycle of 3 survey scans from m/z 380 to m/z 985 and 45 MS2 scans

with precursor isolation width of 13.5 $m/z$ to cover the mass range. DIA raw data files were analyzed by an integrated software suite DIA-NN[112]. The database used in the DIA-NN search (library-free mode) is the latest *A. oryzae* UniProt proteome FASTA sequences plus the protein sequences of the heterologous proteins and common proteomic contaminants. DIA-NN determines mass tolerances automatically based on first-pass analysis of the samples with automated determination of optimal mass accuracies. The retention time extraction window was determined individually for all MS runs analyzed via the automated optimization procedure implemented in DIA-NN. Protein inference was enabled, and the quantification strategy was set to Robust LC = High Accuracy. Output main DIA-NN reports were filtered with a global FDR = 0.01 on both the precursor level and protein group level. The total peak area of tryptic peptides of identified proteins was used to plot the quantity of the targeted proteins in the samples.

### Extraction and LC–MS analysis of ergothioneine and heme in fungal mycelium and reference samples

**Extraction and analysis of heme.** Extraction was performed according to the protocol specified in ref. 113, with minor modifications. Lyophilized fungal biomass was ground into a homogeneous powder using a mortar and pestle. Approximately 30 mg of the powder was then transferred to a bead-beating tube (Lysing Matrix Z, MP Biomedicals, catalog#: 116961050-CF) and 1 mL of TE buffer (10 mM Tris, 1 mM EDTA, pH 8) was added. The tube was vortexed to suspend the powder and was then subjected to bead beating for 2 × 1 min using the Biospec Mini Beadbeater. 750 μL of the bead-beaten solution was then transferred to 15 mL conical tubes containing 4 mL of 8:2 acetonitrile:1.7 M HCl. The tubes were then vortexed for 20 min. Then, 1 mL of saturated 0.25 g MgSO$_4$•7H$_2$O was added to each tube, followed by the addition of 0.1 g NaCl. This created a separation of the aqueous and organic layers. The tubes were then vortexed for 10 min, followed by spinning down at 5000 rcf for 10 min to separate the layers. 100 μL of the top layer was transferred to an LC–MS vial for analysis. In addition to the wild-type and engineered biomass, we extracted heme from lyophilized plant-based ground beef (IMPOSSIBLE Foods Inc, 12 oz IMPOSSIBLE™ burger made from plants).

**LC–MS analysis of heme.** For the LC–MS analysis, analytes were chromatographically separated with a Kinetex XB-C18 column (50-mm length, 2.1-mm internal diameter, 2.6-μm particle size; Phenomenex, Torrance, CA) at 50 °C using a 1260 Infinity HPLC system (Agilent Technologies). The injection volume for each measurement was 5 μL. The mobile phase was composed of 0.1% formic acid in water (as mobile phase A) and 0.1% formic acid in acetonitrile (as mobile phase B). Analytes were separated via the following gradient: linearly increased from 20%B to 45.5%B in 1.7 min, linearly increased from 45.5%B to 95%B in 0.2 min, held at 95%B for 1.6 min, linearly decreased from 95%B to 20%B in 0.2 min, held at 20%B for 1.3 min. A flow rate of 1 mL/min was used throughout. The total run time was 5 min. The HPLC system was coupled to an Agilent Technologies 6520 Quadrupole Time-of-Flight Mass Spectrometer (QTOF-MS) with a 1:4 post-column split. Nitrogen gas was used as both the nebulizing and drying gas to facilitate the production of gas-phase ions. Drying and nebulizing gases were set to 10 L/min and 30 psi, respectively, and a drying gas temperature of 330 °C was used throughout. Fragmentor, skimmer, and OCT1 RF voltages were set to 250 V, 65 V, and 400 V, respectively. Electrospray ionization (ESI) was conducted in the positive-ion mode with a capillary voltage of 3.5 kV. MS experiments were carried out in the full-scan mode ($m/z$ 60–1100) at 0.86 spectra per second for the detection of $[M + H]^+$ ions. The instrument was tuned for a range of $m/z$ 50–1700. Prior to LC-ESI-TOF MS analysis, the TOF MS was calibrated with the Agilent ESI-Low TOF tuning mix. Mass accuracy was maintained via reference ion mass correction, which was performed with purine and HP-0921 (Agilent Technologies). Data acquisition was carried out by MassHunter Workstation Software version B.08.00 (Agilent Technologies). Data processing was carried out by MassHunter Workstation Qualitative Analysis version B.06.00 and MassHunter Quantitative Analysis version 10.00. External calibration curves were used to quantify the analytes. Hemin chloride (Sigma–Aldrich, #3741) was used as the standard. The mass spectrum in the standards and samples corresponded to that from other experimentally validated studies of intracellular heme[114]. Calculated concentrations obtained from LC–MS analysis were normalized to the initial dry sample weight used for extraction.

**Extraction and analysis of ergothioneine.** All samples were lyophilized prior to analysis. For extraction from solid, samples were ground into a fine powder using a mortar and pestle and then approximately 30 mg was transferred to tubes for homogenization (Lysing Matrix Z, MP Biomedicals, catalog#: 116961050-CF). 1 mL of 20% methanol with 0.1% formic acid was added and samples were subjected to bead beating for 2 × 1 min. Following bead-beating, samples were spun down at 12,000 RCF for 10 min to separate the solids. 500 μL supernatant was transferred to a centrifugal spin filter to remove any particulates larger molecules (3 kDa cutoff) (Amicon Ultra, Sigma–Aldrich, Catalog # UFC500324). The flow-through was collected and subjected to analysis by LC–MS. In addition to wild-type and engineered fungal biomass, we extracted ergothioneine from the fruiting body of the oyster mushroom (*Pleurotus ostreatus*). The mushroom was purchased fresh (from Berkeley Bowl in Berkeley, CA) and subjected to lyophilization prior to extraction using the procedure above.

**LC–MS analysis of ergothioneine.** For LC–MS analysis, analytes were chromatographically separated with a Kinetex HILIC column (100-mm length, 4.6-mm internal diameter, 2.6-μm particle size; Phenomenex, Torrance, CA) at 20 °C using a 1260 Infinity HPLC system (Agilent Technologies, Santa Clara, CA, USA). The injection volume for each measurement was 2 μL. The mobile phase was composed of 10 mM ammonium formate (prepared from a pre-made solution from Sigma–Aldrich, St. Louis, MO, USA) and 0.2% formic acid (from an original stock at ≥98% chemical purity from Sigma–Aldrich) in water (as mobile phase A) and 10 mM ammonium formate and 0.2% formic acid in 90% acetonitrile with the remaining solvent being water (as mobile phase B). The solvents used were of LC–MS grade and purchased from Honeywell Burdick & Jackson, CA, USA. Analytes were separated via the following gradient: linearly decreased from 90%B to 70%B in 4 min, held at 70%B for 1.5 min, linearly decreased from 70%B to 40%B in 0.5 min, held at 40%B for 2.5 min, linearly increased from 40%B to 90%B in 0.5 min, held at 90%B for 2 min. The flow rate was changed as follows: 0.6 mL/min for 6.5 min, linearly increased from 0.6 mL/min to 1 mL/min for 0.5 min, held at 1 mL/min for 4 min. The total run time was 11 min. The HPLC system was coupled to an Agilent Technologies 6520 Quadrupole Time-of-Flight Mass Spectrometer (QTOF-MS) with a 1:4 post-column split. Nitrogen gas was used as both the nebulizing and drying gas to facilitate the production of gas-phase ions. Drying and nebulizing gases were set to 12 L/min and 25 psi, respectively, and a drying gas temperature of 350 °C was used throughout. Fragmentor, skimmer, and OCT1 RF voltages were set to 100 V, 50 V, and 250 V, respectively. Electrospray ionization (ESI) was conducted in the positive-ion mode with a capillary voltage of 3.5 kV. MS experiments were carried out in the full-scan mode ($m/z$ 70–1100) at 0.86 spectra per second for the detection of $[M + H]^+$ ions. The instrument was tuned for a range of $m/z$ 50–1700. Prior to LC-ESI-TOF MS analysis, the TOF MS was calibrated with the Agilent ESI-Low TOF tuning mix. Mass accuracy was maintained via reference ion mass correction, which was performed with purine and HP-0921 (Agilent Technologies). Data acquisition was carried out by MassHunter Workstation Software version B.08.00 (Agilent Technologies). Data processing was carried out by MassHunter Workstation Qualitative

Analysis version B.06.00 and MassHunter Quantitative Analysis version 10.00. External calibration curves were used to quantify the analytes. Ergothioneine (Sigma–Aldrich, #7521-25MG) was used as the standard. The mass spectrum in the standards and samples corresponded to that from other experimentally validated studies of intracellular ergothioneine[115]. Calculated concentrations obtained from LC–MS analysis were normalized to the initial dry sample weight used for extraction.

### Protein and amino acid analysis in fungal mycelium

Protein content was analyzed by combustion method, directly following the *Method 990.03* described in ref. 116. The combustion was performed using Leco FP-528 Nitrogen Combustion Analyzer (Leco). Crude Protein was calculated as Nitrogen × 6.25. Amino acid composition was analyzed using acid hydrolysis of lyophilized fungal biomass, directly following the protocols of *Method 994.12* described in ref. 116, *Method 982.30* in ref. 117, as well as the methods described in ref. 118.

### Statistics and Reproducibility

No statistical method was used to predetermine sample size. $n = 3$ was chosen as the minimal number of replicates for experimental characterization. We determined this to be sufficient based on internal controls (using previously characterized promoters and fluorescent genes) to capture biological variability between transformants/strains. All microscopy images and PCR results for confirming insertion/excision displayed are representative of at least three biological replicates. No data were excluded from the analyses. The experiments were not randomized. The investigators were not blinded to allocation during experiments and outcome assessment.

### Reporting summary

Further information on research design is available in the Nature Portfolio Reporting Summary linked to this article.

## Data availability

The authors declare that all data supporting the findings of this study are available within the paper, supplementary information, the supplementary data files, and in the source data file. Source data are provided as a Source Data file. Strains and plasmids (and their associated sequences) generated in this study have been deposited in the JBEI Public Registry (https://public-registry.jbei.org/). See Supplementary Tables 11–12 for plasmids and strain information. Outputs of computational analysis for identification of candidate endogenous neutral loci and bidirectional promoters are available in Supplementary data files 1 and 2. Output of mass spectrometry data are available as source data. The generated mass spectrometry proteomics data have been deposited to the ProteomeXchange Consortium via the PRIDE[94] partner repository with the dataset identifier "PXD043152". The genome sequences of the *A. oryzae* strains obtained from NRRL and sequenced as part of this study have been deposited to the Sequence Read Archive under Bioproject "PRJNA987873". The previously published transcriptome data used to identify endogenous neutral loci and bidirectional promoters in *Aspergillus oryzae* can be found in GenBank under BioProject accession: "PRJDB8293". CAoGD (Comprehensive *Aspergillus oryzae* Genome Database) v.2.4 was used to identify the genomic location and sequence identity of endogenous promoters and genes targeted in transformations (https://nribf21.nrib.go.jp/CAoGD/). Source data are provided with this paper.

## Code availability

No custom code was used in the analysis of data, and all the previously published software used as well as the relevant commands has been cited in the methods. The automatic phylogenomic analysis from the core genomes of *Aspergillus oryzae* strains was executed using the script available at https://github.com/WeMakeMolecules/Core-to-Tree. DIA-NN is freely available for download from https://github.com/vdemichev/DiaNN.

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

## Acknowledgements

V.M.R. was supported by the Miller Institute at the University of California, Berkeley. P.C.M. and J.D.K. were supported by Novo Nordisk Foundation grant no. NNF20CC0035580. C.V.D.L. was supported by Novo Nordisk Foundation grant NNF21OC0065495. This work was part of the DOE Joint BioEnergy Institute (https://www.jbei.org) supported by the U.S. Department of Energy, Office of Science, Office of Biological and Environmental Research under contract DE-AC02-05CH11231 between Lawrence Berkeley National Laboratory and the U.S. Department of Energy.

## Author contributions

VMR and JDK conceptualized the study. VMR developed and optimized transformation methods, established the neutral loci, endogenous promoters, and vectors for transformation, and engineered the final strains. CVDL evaluated parts for the Synthetic expression system and conducted transformation of A. oryzae NRRL strains. PCM conducted bioinformatics analysis, including genome annotation and assembly, phylogenetic analysis, and identification of endogenous neutral loci and bidirectional promoters. YC and CJP conducted proteomics experiments and analysis. RK and EEKB conducted targeted LC–MS experiments and analysis. All authors approved the final manuscript.

## Competing interests

J.D.K. has financial interests in Amyris, Ansa Biotechnologies, Apertor Pharma, Berkeley Yeast, Cyklos Materials, Demetrix, Lygos, Napigen, ResVita Bio, and Zero Acre Farms. V.M.R. and J.D.K. are listed as inventors on a provisional patent (US22/42816) which relates to the methods composition described in the engineering of heme metabolism in edible fungal biomass. The other authors declare no competing interests.
