## [Peer Review File · Nature Communications]

REVIEWER COMMENTS

Reviewer #1 (Remarks to the Author):

The aim of the study described in this Nat Comms Article was to develop a “comprehensive genetic toolkit that includes an RNP-based method for genome modification, neutral loci, and new promoter tools”. In addition, the authors provided two proof of concept using the developed toolkit to enhance ergothioneine and the eight step heme biosynthetic pathway as a use case for enhancing nutritional and sensory properties of *A. oryzae*. Outcomes from this study would expand available synthetic biology toolkit for fungal food production.

Noteworthy findings include:

(1) recyclable RNP-based CRISPR-Cas9 by targeting the *wA* locus which controls spore pigmentation, *wA* locus targeting and function in RIB40 in addition to 5 other evolutionary and application related *Aspergillus* spp. and integration efficiency independent of arm length (marker recycling was also confirmed using *wA:gfp* and *niaD::mCherry*)

(2) systematic and comprehensive identification and evaluation of neutral insertion loci using computational and experimental methods; all loci displayed higher expression than the *amyA* positive control; IE and magnitude of GFP expression were largely consistent

(3) Use of Synthetic Expression System (SES) using a previously characterized Bm3R1-VP16 sTF to drive mCherry expression; the strongest SES core promoter, *Ao_0583* in this study is ~6-fold stronger than the commonly used, strong, starch-inducible promoter *pAmyB*; SES permits high protein expression independent of carbon source, glucose and dextrin

(4) Expression and production of (i) amino acid ergothioneine encoded by two enzymes, *Egt1* and *2*; two strains in *A. oryzae* were generated VWR-Eg1-2 (one copy) and VWR-Eg1_2 (two copies) driven by endogenous bidirectional promoters

(ii) heme generating strain VMR-HEM-v1; 4-fold higher heme production compared to control RIB40

Strengths: I enjoyed reading the article, it was well written with clear objectives of the study, well thought out experimental designs, and systematic development and testing of toolkit. Supplemental material also provided detailed support of the toolkit development.

Minor detailed comments below:

Introduction:

L87 1) not necessary here

Results:

L147 ... should say Table S2?

L157 Fig 1D 90% IE with 950 bp homology arms – how many transformants were screened at each length – authors should include this since % depends on # of transformations, and frequency of positive transformants; OK – methods L290-291

L387-388 The final strain contained a total of six separate modifications; I may have missed this but I did not find methods describing construction of this strain

L411 figure 4E should be cited in text

L413 ...that the

Fig 2D all loci except chro4_1; should this say 3_1 (L542) based on the graph 4_1 displayed 50% IE, also L223-224 in text describes 3_1

Supplemental information:

Table S1 were the same sgRNA as in Table 1 for RIB40 used for diverse strains

Fig S1 how was efficiency calculated? L19-20

Fig S2 L49 how many days of growth at what conditions – did any of the integration phenotypically impacted growth (colony diameter) OK – methods L240 under growth conditions, L285-286

Fig S3 panel E missing description in figure legend

Table S4 and Fig S5; table S4 missing information for plasmid targeting chro4_2

Fig S9 authors provided sequence alignment of Egt1 and Egt2 for *N. crassa* and *A. oryzae*; could authors provide a schematic for the differences in neutral loci insertion, bidirectional promoter and main differences in approach for generation of two strains in Fig 4B; I also did not find methods that describe generation of these strains

-could authors include retention time and absolute peak intensity peak for ergo (standard from Sigma was used) via LC-MS to allow replication of methods, citation of previous method/protocol from literature (I see citation ref 27 for heme but not ergo?)

Fig S10 why is figure cited in L184

Fig S11 please include error bars for technical and biological replicates for protein analysis; also why were heme strain VMR_HEM_v1 analyzed for biomass, protein levels and amino acid profiling but not ergo strains, Eg1-2 and Eg1_2?

-one replicate from each strain; how would authors know reliability and levels of production of ergo and heme from one replicate?

- I also did not find methods that describe generation of these strains

-could authors include retention time of peak for heme (hemin chloride was used as a standard) via LC-MS to allow replication of methods

Methods:

L359 superscript 1.2×10^7

Reviewer #2 (Remarks to the Author):

The authors developed genetic tools with CRISPR-Cas9 in the edible filamentous fungus *Aspergillus oryzae*. The application of such technologies to fungal meat alternatives will provide great impacts to future studies. The authors developed genetic tools including neutral loci for gene integration and new promoters by employing the CRISPR-Cas9 method. However, the contributions of the genetic tools to the production of food alternatives seem to be partial or limited. More improvements of the technologies are still required to fully support their conclusions, and there are many unclear points in the manuscript due to insufficient explanation and writing mistakes.

Major points

1. Throughout the Introduction and Discussion sections, the authors focused on food alternatives rather than development of new genetic tools. It seems that the authors combined previous genetic techniques involving CRISPR-Cas9 Ribonucleoprotein complexes (RNPs), which have already been reported in filamentous fungi (Abdallah et al., *mSphere*. 2017;2:e00446-17 and following papers). They should describe about the issues of advantages of the new genetic tools although it is possible to construct the strains by using the previous techniques.

2. As the strategy to construct the strains used in Figure 4 was not described well, it is difficult to evaluate the contribution of their genetic tools to the strain construction.

The endogenous bidirectional promoter p4_2 was less effective than the mono-directional promoter, pTEF1 (Figure 4B). Finally, the bidirectional promoters were not used for heme production requiring 6 genes, and the artificial promoters were limitedly used (Figure 4C). For application of the bidirectional promoters, more improvements such as enhanced expression of multiple genes are required.

3. It is not clear why two RNPs were used. Use of single RNP is probably sufficient to allow DNA integration, and two RNPs may increase off-target mutations and costs for genetic modification. Did two RNPs show a higher modification efficiency than single RNP?

To generate Δ pyrG stains, two RNP complexes were used without any template DNA. However, the authors only confirmed auxotrophies but did not investigate the deletion pattern of pyrG. Therefore, unexpected deletion of the flanking genes may occur. As the efficiency to generate Δ pyrG was not examined, it is not clear whether the generation of Δ pyrG was performed "easily" (Line 152).

In addition, the wA mutants in NRRL strains were not investigated at a genetic level. It is not clear whether these mutants contain an expected deletion, which should be confirmed experimentally.

4. In Figures 1D, S1F, and S4D, the graphs do not contain error bars, and the number of experimental trials was not written. As it is difficult to evaluate the reproducibility, these results do not seem to be enough to support the authors' conclusions.

Similarly, in Lines 354-357 and Figure 4B, it seems to be difficult to conclude that the ergothioneine level of VMR-EG1-2 was higher than that of oyster mushroom without statistic tests.

5. The PCR results shown in figures are difficult to be precisely evaluated, especially > 4 kb, due to the gel with a high concentration of agarose. The expected band sizes were roughly described like ~4 kb and ~750 bp (Figures 1, S1, and S3). Furthermore, the strategies to construct *yA* and *niaD* mutants are not shown in Figure S3. The authors should show the construction strategies and describe the results precisely.

6. GFP fluorescence was measured only with conidia in Figure 2. Considering the aim of this study, gene expression level in mycelia should be examined.

7. Only conidia formation was measured to show the fungal growth in Figure S6. To indicate the neutral loci, growth phenotypes should be shown.

Minor points

Line 87: "1)" should be deleted.

Lines 113, 169, 268, 294, 315, and 422: The numbers indicating references should be upright (not italic).

Line 142: "locus" should be upright (not italic).

Lines 143, 148, 152, 521, and Figure S2: " Δ " should be upright (not italic).

Lines 145-150: As "a diverse group of *A. oryzae*" is based on the whole-genome sequencing analyses, the description about whole-genome sequencing should be placed in the former sentence, and then the evaluation of the sequencing results should be described.

Lines 147 and 149-150: Table S1 > Table S2

Line 165: Figure S1 > Figure S1F

Lines 187-189: The conclusion here seems to be repeated with that mentioned in Lines 175-176. The authors should clarify these conclusions.

Lines 189-191: It is not clear how this method "complements" the previous technique.

Line 223: Figure 2C > Figure 2D. Then, no explanation about Figure 2C exists in the main text.

Line 223: The sentence "9 loci showed high-efficiency integration (>50%)" means that the 9 loci include *chro4_1* but not *chro3_1*. In the Figure 2 legend (Lines 542-543), however, the authors mentioned "all loci except for *chro4_1* displayed a high efficiency of integration (>50%)". Which is correct?

Line 224 and Figure 2 legend: As names of the intergenic regions were in upright in Line 224, those in Figure 2 legend should be italic.

Lines 226 and 228: Figure 2D > Figure 2E

Lines 227-228: *chro6-1* and *chro7-1* > *chro6_1* and *chro7_1* (not hyphen but underbar)

Line 304: Why were the core promoters of *gpdA* and *hhfA* but not of *Ao_0583* used for construction of bidirectional promoter?

Lines 304 and 567: *gpdA* and *hhfA* should be italic.

Line 306: Figure 1F > Figure 3F

Line 321: "*oryzae*" should be italic.

Lines 351 and 583: "their own strong promoter" and "their own promoter" are confusing. "their own promoters" may mean those of *AO_Egt1* and *AO_Egt2* genes (but not *pTEF1*).

Line 353: Does "co-expression" mean the expression from the bidirectional promoter? Dual gene integration also results in "co-expression".

Line 386: It is not clear why Soy Leghemoglobin (= LegH?) was expressed.

Line 406 and Figure 4: Figure 4E > Figure 4D

Line 525: Unify the description of mCherry gene name. "mCherry"(Line 180) or "mcherry" (Line 525).

Line 782: What is the reference 64 "Administration, F. a. D. (2021)."?

Supplementary Lines 16-19: In Fig. S1 legends, "D)" and "E)" should be reordered.

Supplementary Lines 29 and 83: *Ca9* > *Cas9*

Supplementary Line 58: B) Successful > E) Successful

Supplementary Lines 113, 240, 272, 304, 340, and 459: *Aspergillus oryzae* > *A. oryzae*

Supplementary Line 128: "Candidate(s)" was repeated.

Supplementary Line 182: Does leghemoglobin LegH really drive the hem biosynthesis flux?

Supplementary Lines 183-184: In Fig. S10 legend, the sentence here was just diverted from the main text (Lines 384-385).

Supplementary Lines 207-209: In Table S9, clarify the upper and lower characters of primer sequences.

Supplementary Lines 207-209: In Table S9, not all the primers used in Figures 1 and S1 may be included.

Supplementary Lines 214-215: In the Description column of Table S11, the names of the ergothioneine biosynthesis genes EgtA and EgtB are different from those (AO_Egt1 and AO_Egt2) designated by the authors.

Supplementary Lines 214-215: In Table S11, why did the authors increase the LegH expression cassette by using the artificial promoters together with Bm3R1_VP16? Was the copy number increase effective for heme production?

Supplementary Line 226: Supplementary table S9 > Supplementary Table S9

Supplementary Lines 246-247: grams > g, MgSO₄·7H₂O > MgSO₄·7H₂O (4 and 2 in subscript), KH₂PO₄ > KH₂PO₄ (2 and 4 in subscript)

Supplementary Line 264: MgSO₄x7H₂O > MgSO₄·7H₂O

Supplementary Line 359: 1.2*10⁷ > 1.2*10⁷ (7 in superscript)

Supplementary Line 395: "1)" may not be necessary.

Supplementary Line 504: "genome" should be in upright (not italic).

Supplementary data S1: Although the occurrence values were shown, it is not clear how the highly expressed genes were determined. Was a threshold value of read counts determined?

Scale bars are missing in the microscopic images.

Reviewer #3 (Remarks to the Author):

L37, not sure about this statement, there are other technologies/platforms causing the same issues, and this statement is only blaming food industry. I would suggest that authors reformat the statement.

Introduction: Could authors please add more information about 1- GMO, 2- Acceptability, 3-possible food safety concerns?

Did author measure the color of the final product?

I did not see the quality of the heme compared to the conventional heme in response to the thermal processing/cooking.

The approach in this study is interesting, however, since the final product tends to be eaten, it should mimic the conventional meat.

Could author also provide secondary structure information about the heme? This could be done by FTIR, and Chemometrics.

REVIEWER COMMENTS

Reviewer #1 (Remarks to the Author):

The aim of the study described in this Nat Comms Article was to develop a “comprehensive genetic toolkit that includes an RNP-based method for genome modification, neutral loci, and new promoter tools”. In addition, the authors provided two proof of concept using the developed toolkit to enhance ergothioneine and the eight step heme biosynthetic pathway as a use case for enhancing nutritional and sensory properties of *A. oryzae*. Outcomes from this study would expand available synthetic biology toolkit for fungal food production.

Noteworthy findings include:

- (1) recyclable RNP-based CRISPR-Cas9 by targeting the wA locus which controls spore pigmentation, wA locus targeting and function in RIB40 in addition to 5 other evolutionary and application related *Aspergillus* spp. and integration efficiency independent of arm length (marker recycling was also confirmed using wA:gfp and niaD::mCherry)
- (2) systematic and comprehensive identification and evaluation of neutral insertion loci using computational and experimental methods; all loci displayed higher expression than the amyA positive control; IE and magnitude of GFP expression were largely consistent
- (3) Use of Synthetic Expression System (SES) using a previously characterized Bm3R1-VP16 sTF to drive mCherry expression; the strongest SES core promoter, Ao_0583 in this study is ~6-fold stronger than the commonly used, strong, starch-inducible promoter pAmyB; SES permits high protein expression independent of carbon source, glucose and dextrin
- (4) Expression and production of (i) amino acid ergothioneine encoded by two enzymes, Egt1 and 2; two strains in *A. oryzae* were generated VWR-Eg1-2 (one copy) and VWR-Eg1_2 (two copies) driven by endogenous bidirectional promoters
(ii) heme generating strain VMR-HEM-v1; 4-fold higher heme production compared to control RIB40

Strengths: I enjoyed reading the article, it was well written with clear objectives of the study, well thought out experimental designs, and systematic development and testing of toolkit. Supplemental material also provided detailed support of the toolkit development.

We thank the reviewer for highlighting the major findings and potential impact of our work.

Minor detailed comments below:

Introduction:

L87 1) not necessary here

We have edited this typo.

Results:

L147 ... should say Table S2?

We thank the reviewer for pointing out this reference. We referenced Table S1 to show the gene identifier and the associated sgRNAs used in the development of the RNP-based editing method in *A. oryzae* RIB40. We have edited the title and caption of Table S1 to clarify this point.

L157 Fig 1D 90% IE with 950 bp homology arms – how many transformants were screened at each length – authors should include this since % depends on # of transformations, and frequency of positive transformants; OK – methods L290-291

We thank the reviewer for raising this point. The reviewer is correct that the % depends on the number of screened transformants. Although the methodology is described in the methods section, we have updated the figure and the associated caption to better represent the data. Throughout the manuscript, a bar graph has been replaced with a table showing how many transformants were screened at condition and the frequency of transformants identified to harbor insertion at the locus of interest. This representation is directly comparable to how editing efficiency was presented in the previous publication of the plasmid-based CRISPR-Cas9 method for gene editing in *Aspergillus oryzae* (Katayama et al, 2019), as well as in other publications describing gene editing in filamentous fungi (Liu et al, 2015; Pohl et al, 2016).

L387-388 The final strain contained a total of six separate modifications; I may have missed this but I did not find methods describing construction of this strain.

We thank the reviewer for highlighting this point. We have addressed this in several ways. First, the engineering strategy is more clearly described in the main text (lines 553-580 for ergothioneine, lines 613-648 for heme). Additionally, we have visually represented the engineering strategies for ergothioneine and heme in the supplementary information (Fig. S11 and S14, respectively). We have also described in the methods more clearly how the final strains listed in Table S12 were generated (lines 691 – 701).

L411 figure 4E should be cited in text

Thanks to the reviewer, we noticed a mistake in which the original Figure 4 contained a panel 4E instead of 4D. We have edited the figure and the text accordingly.

L413 ...that the

We have edited this typo.

Fig 2D all loci except chro4_1; should this say 3_1 (L542) based on the graph 4_1 displayed 50% IE, also L223-224 in text describes 3_1

We thank the reviewer for pointing out this typo. We have corrected the caption and text accordingly.

Supplemental information:

Table S1 were the same sgRNA as in Table 1 for RIB40 used for diverse strains

The reviewer is correct that the same sgRNAs were used in the editing of both the RIB40 and the five different NRRL strains. We have edited the caption of Table S1 accordingly to highlight this point.

Fig S1 how was efficiency calculated? L19-20

We thank the reviewer for pointing out this missing information. We have edited the caption to include more information about how efficiency was calculated throughout the manuscript (see response above).

Fig S2 L49 how many days of growth at what conditions – did any of the integration phenotypically impacted growth (colony diameter) OK – methods L240 under growth conditions, L285-286

We thank the reviewer for pointing out this information which was missing from the figure caption. We have updated it accordingly. Additionally, we want to highlight to the reviewer that we did not observe any gross effects of *pyrG* mutation on colony growth across the strains shown in Figure S3. Finally, based on reviewer feedback, we have conducted additional analysis to calculate the efficiency of *pyrG* mutation and characterize the genomic changes associated with the RNP-targeting across the strains (Fig. S3).

Fig S3 panel E missing description in figure legend

We thank the reviewer for pointing out this typo. We have edited accordingly. The updated Figure is now Figure S4. Additionally, we have edited Figure S4 to more clearly show the integration strategy, efficiencies, and how the insertion / loop-out corresponds to the PCR amplicons displayed in gels.

Table S4 and Fig S5; table S4 missing information for plasmid targeting chro4_2

We thank the reviewer for pointing out this missing information. The plasmids for chro3_1 and chro4_2 were excluded from the final, experimentally validated, publicly available toolkit as they did not show any evidence of integration by PCR (chro3-1) or did not express GFP from the locus (chro4_2) and thus we feel would not prove useful to the scientific community for further engineering efforts. We have highlighted this in the caption of Table S5.

Fig S9 authors provided sequence alignment of Egt1 and Egt2 for *N. crassa* and *A. oryzae*; could authors provide a schematic for the differences in neutral loci insertion,

bidirectional promoter and main differences in approach for generation of two strains in Fig 4B; I also did not find methods that describe generation of these strains

We thank the reviewer for raising this issue. As detailed above, we have addressed this lack of information in several ways. First, the engineering strategy is more clearly described in the main text (lines 553-580 for ergothioneine, lines 613-648 for heme). Additionally, we have visually represented the engineering strategies for ergothioneine and heme in the supplementary information (Fig. S11 and S14, respectively). We have also described in the methods more clearly how the final strains listed in Table S12 were generated (lines 691 – 701).

-could authors include retention time and absolute peak intensity peak for ergo (standard from Sigma was used) via LC-MS to allow replication of methods, citation of previous method/protocol from literature (I see citation ref 27 for heme but not ergo?)

We thank the reviewer for pointing out this missing information. References have now been added to the LC-MS methods sections describing both ergothioneine and heme detection. In addition, based on the reviewer feedback, have added retention time and mass fragmentation patterns for the detection of both ergothioneine (Fig. S12) and heme (Fig. S15) across the standard chemical and the engineered strains and foods for comparison. These data lend further confidence to the outcomes of our engineering efforts, and we anticipate that the inclusion of these data this will facilitate replication in future studies.

Fig S10 why is figure cited in L184

We have removed this typo.

Fig S11 please include error bars for technical and biological replicates for protein analysis; also why were heme strain VMR_HEM_v1 analyzed for biomass, protein levels and amino acid profiling but not ergo strains, Eg1-2 and Eg1_2?

-one replicate from each strain; how would authors know reliability and levels of production of ergo and heme from one replicate?

We thank the reviewer for this thoughtful comment. All metabolite analysis was conducted in biological triplicate (Figure 4B and 4C). However, the amino acid and protein analysis was initially only conducted with a single strain due to the large amount of dry material (>10 grams) needed to conduct the analyses. Additionally, in the initial study, the heme strain was considered the most promising for potential downstream food applications due to its compositional and visual similarity to a widely available plant-based meat alternative incorporating heme for flavor and color. As a result, only this strain was included in the protein and amino acid analysis. Protein and amino acid content were analyzed from a single sample as a preliminary investigation into the nutritional profile. However, in response to the reviewer comment, in the revised manuscript, we have included protein

content in triplicate for both the heme (Fig. S16) and ergothioneine strains (Fig. S13). We have also conducted the amino acid analysis in triplicate for the heme strain (Fig. S16). We have changed the figure and its associated caption accordingly.

- I also did not find methods that describe generation of these strains

See response above regarding more detailed information about engineering strategy and strain construction.

-could authors include retention time of peak for heme (hemin chloride was used as a standard) via LC-MS to allow replication of methods

The retention time and mass spectrum for heme are now clearly described in Fig. 15 and have been included in the methods (LC-MS section, lines 1042-1135). We want to point out that in solution, hemin chloride dissociates into heme, and as can be seen in the data, the major analyte in detected is the same as in previous LC-MS studies analyzing heme in biological samples (Sana et al, 2008). We have added the relevant reference and clarified this point in the caption of Fig. S15.

Methods:

L359 superscript 1.2×10^7

We have corrected typos related to superscript throughout the manuscript.

References:

Katayama T, Nakamura H, Zhang Y, Pascal A, Fujii W, Maruyama JI. Forced Recycling of an AMA1-Based Genome-Editing Plasmid Allows for Efficient Multiple Gene Deletion/Integration in the Industrial Filamentous Fungus *Aspergillus oryzae*. *Appl Environ Microbiol*. 2019 Jan 23;85(3):e01896-18. doi: 10.1128/AEM.01896-18. PMID: 30478227; PMCID: PMC6344613.

Liu R, Chen L, Jiang Y, Zhou Z, Zou G. Efficient genome editing in filamentous fungus *Trichoderma reesei* using the CRISPR/Cas9 system. *Cell Discov*. 2015 May 12;1:15007. doi: 10.1038/celldisc.2015.7. PMID: 27462408; PMCID: PMC4860831.

Pohl C, Kiel JA, Driessen AJ, Bovenberg RA, Nygård Y. CRISPR/Cas9 Based Genome Editing of *Penicillium chrysogenum*. *ACS Synth Biol*. 2016 Jul 15;5(7):754-64. doi: 10.1021/acssynbio.6b00082. Epub 2016 May 3. PMID: 27072635.

Sana TR, Waddell K, Fischer SM. A sample extraction and chromatographic strategy for increasing LC/MS detection coverage of the erythrocyte metabolome. *J Chromatogr B Analyt Technol Biomed Life Sci*. 2008 Aug 15;871(2):314-21. doi: 10.1016/j.jchromb.2008.04.030. Epub 2008 Apr 26. PMID: 18495560.

Reviewer #2 (Remarks to the Author):

The authors developed genetic tools with CRISPR-Cas9 in the edible filamentous fungus *Aspergillus oryzae*. The application of such technologies to fungal meat alternatives will provide great impacts to future studies. The authors developed genetic tools including neutral loci for gene integration and new promoters by employing the CRISPR-Cas9 method. However, the contributions of the genetic tools to the production of food alternatives seem to be partial or limited. More improvements of the technologies are still required to fully support their conclusions, and there are many unclear points in the manuscript due to insufficient explanation and writing mistakes.

We thank the reviewer for recognizing the great potential of our work for food engineering and appreciate the thoughtful and detailed comments on specific approaches and experiments detailed in the manuscript. We are confident that the additional experimental characterization and writing clarification suggested by the reviewer underlying technology have strengthened the paper and further highlighted the utility of our tools. However, we disagree with the reviewer's opinion that the contribution of the genetic tools to the production of food alternatives was partial or limited. To our knowledge, this is the first example of genetic modification of edible fungal biomass for enhanced nutritional value and sensory appeal. As detailed below, this proof-of-concept was made possible by the development of the genetic toolkit.

Major points

1. Throughout the Introduction and Discussion sections, the authors focused on food alternatives rather than development of new genetic tools. It seems that the authors combined previous genetic techniques involving CRISPR-Cas9 Ribonucleoprotein complexes (RNPs), which have already been reported in filamentous fungi (Abdallah et al., mSphere. 2017;2:e00446-17 and following papers). They should describe about the issues of advantages of the new genetic tools although it is possible to construct the strains by using the previous techniques.

We agree with the reviewer on this point, as the Introduction and Discussion sections focus mainly on the urgent challenges of reforming the food system and the potential of filamentous fungi for developing more sustainable methods of food production. The challenge to develop meat alternatives is of great societal importance that served as the major motivation for our work, and we feel that the major novelty and societal impact of our work lies in the application of bioengineering to alter the properties of edible fungal mycelium. However, our work involves the development of genetic tools that have broad applications in food as well as other fields. While we have chosen not to focus on these genetic tools as the major motivation and impact of our work, we nonetheless believe that this aspect of our manuscript presents many new and promising engineering opportunities for the field.

The reviewer correctly describes that RNP complexes have been used for gene editing in filamentous fungi for a few years, and we regret that the manuscript may, as written, may have overlooked this important fact. We have now highlighted the existence of RNP editing in fungi and cited relevant sources across diverse fungal hosts in the introduction (Line 168-170) and discussion (lines 707-709). We have also emphasized that this approach had not been experimentally characterized and established for gene integration in *A. oryzae* at the start of our study. Thus, rather than broadly presenting RNP-based gene editing as a new approach to gene editing among all filamentous fungi, the major impact on genetic engineering lies in the creation of an experimentally validated, publicly available toolkit for gene integration and expression that is, to our knowledge, unique among molds and mushrooms. Together with the method for RNP editing, these efforts have vastly expanded the number of characterized DNA parts available for engineering *A. oryzae*. For example, the characterization of the synthetic expression system and the discovery of new promoters now enables-medium independent protein expression that is stronger than the gold standard strongest promoter (AmyB) in *A. oryzae* (Fig. 3). Additionally, the establishment of bidirectional promoters enable expression of multiple genes from single constructs (Figs. 3 and 4). Finally, the identification and establishment of neutral loci presents “landing pads” for precise genome integration, which have previously not been extensively established and evaluated among filamentous fungi (Fig. 2). Additionally, beyond the specific case studies highlighted in the study, we feel that our general approaches – RNP-based integration with locus-specific marker recycling, computational identification and evaluation of neutral loci and new promoters – could inform similar engineering efforts and tool development in other hosts. Tools like ours have revolutionized the engineering of commonly used laboratory hosts such as *S. cerevisiae* (Reider et al, 2017), and we expect a similar impact of this work for *A. oryzae*.

Based on these points, we disagree with the reviewer’s opinion that our strains could have been constructed using previous techniques as their construction involved new elements and approaches. For example, new, experimentally validated neutral loci were used to construct all strains, and the strain construction also involved the new promoters established in the study. We have now more clearly emphasized the novelty and anticipated impact of the toolkit in the discussion section, as well as how our tools were used to enable the engineering for food applications (lines 695-726).

2. As the strategy to construct the strains used in Figure 4 was not described well, it is difficult to evaluate the contribution of their genetic tools to the strain construction. The endogenous bidirectional promoter p4_2 was less effective than the mono-directional promoter, pTEF1 (Figure 4B).

We thank the reviewer for raising this issue. Based on the reviewer’s feedback, we have addressed this issue information in several ways. First, the engineering strategy and rationale is more clearly described in the main text (lines 553-580 for ergothioneine, lines 613-648 for heme). Additionally, we have visually represented the engineering strategies for ergothioneine and heme in the supplementary information (Fig. S11 and S14,

respectively). We have also described in the methods more clearly how the final strains listed in Table S12 were generated (lines 691 – 701).

Regarding the application of the bidirectional promoter p4-2, we disagree with the statement that it was “less effective” than the endogenous pTEF1 mono-directional promoter. While it is true that the promoter was weaker than pTEF1, “less effective” is a matter of definition. In the field of promoter development and synthetic biology, and in our study, the goal is not to develop the strongest promoter. Rather, in this study, we sought to identify new DNA parts and modes of gene regulation. Our study has expanded the number of DNA parts available for gene regulation, and the application of the bidirectional promoter in the ergothioneine strain establishes for the first time the use of such a promoter for metabolite production in *A. oryzae*.

Finally, the bidirectional promoters were not used for heme production requiring 6 genes, and the artificial promoters were limitedly used (Figure 4C). For application of the bidirectional promoters, more improvements such as enhanced expression of multiple genes are required.

We disagree that further bidirectional proof of concept is needed because it would not alter the conclusion of the study or the utility of the toolkit for the scientific community. We feel that, across the heme and ergothioneine engineering efforts, we have provided sufficient proof-of-concept for the application of genetic tools to bioengineered fungal food production. By providing the experimentally validated toolkit to the public, it is our hope that researchers will evaluate the impact and utility of the diverse DNA parts, including the bidirectional promoters, across different applications and engineering challenges in the future.

3. It is not clear why two RNPs were used. Use of single RNP is probably sufficient to allow DNA integration, and two RNPs may increase off-target mutations and costs for genetic modification. Did two RNPs show a higher modification efficiency than single RNP?

We thank the reviewer for this comment. We did not describe the rationale behind choosing two sgRNAs for a single locus in the manuscript. Two RNPs were chosen to increase the likelihood of double-stranded breaks and because the locus-specific marker recycling method established in *A. niger* for plasmid-based CRISPR-Cas9 editing, upon which our approach is based, used two sgRNAs for an individual locus for gene integration (Leynaud-Kieffer et al, 2019). The sgRNA targeting sites, ~300bp from the 5' start of the locus of interest, and ~1000 bp from the 3' end of the locus of interest, were chosen based on the success of gene integration and marker recycling demonstrated in the previous study. A similar dual-sgRNA targeting strategy had also been established in other fungal hosts (Rantasalo et al, 2019; Liu et al, 2020; Gao et al, 2018). We have edited the text to describe the rationale for selecting two targeting sites. While the dual targeting approach displayed high integration efficiency across diverse loci and strains of

A. oryzae, we feel it was important based on the reviewer feedback to evaluate the difference between using one and two RNPs. Therefore, we conducted an additional experiment, utilizing either two RNPs or a single RNP (the 5' or the 3' targeting sequence, as seen in Table S1), to evaluate the impact of integration efficiency across two independent loci (*wA* and *niaD*) in *A. oryzae* RIB40. We observed no major difference in integration efficiency between two or one RNP complex, suggesting that the RNP based integration is highly efficient independent of the specific locus or targeting sequence (main text lines 238-241, Table S3). We have now included this experimental information in the manuscript as we feel it would be informative for researchers attempting RNP-based gene integration in *A. oryzae* and beyond. We cannot rule out that utilizing two RNPs may increase costs for genetic modification and potentially off-target mutations. However, the sgRNAs were designed using the CRISPOR software (<http://crispor.tefor.net/crispor.py>) to select candidates with no predicted off-target mutations in the genome. Additionally, as detailed below, we did not observe any off-target effects on surrounding genes in our mutation of *pyrG* across diverse *A. oryzae* NRRL strains.

To generate Δ *pyrG* stains, two RNP complexes were used without any template DNA. However, the authors only confirmed auxotrophies but did not investigate the deletion pattern of *pyrG*. Therefore, unexpected deletion of the flanking genes may occur. As the efficiency to generate Δ *pyrG* was not examined, it is not clear whether the generation of Δ *pyrG* was performed "easily" (Line 152).

In the initial description of the *pyrG* mutant generation across the NRRL strains, we did not include information about the number of colonies evaluated for uridine/uracil auxotrophies across the strain collection and the number of colonies displaying the expected phenotypes. One of these colonies was used for further transformation and characterization. We have now included this information in Fig. S3. As all analyzed colonies across all strains displayed the expected phenotype, we have changed the word "easily" with "readily" in the main text (line 226) as this more accurately captures the procedure and experimental outcome for generating *pyrG* mutants in the NRRL strains. Additionally, based on the reviewer feedback, we have examined the deletion pattern of *pyrG* across these strains. These analyses (now added to Fig. S3) display a variety of disruptions in *pyrG* adjacent to the predicted cutsites of the RNPs, from a clean deletion to random insertion likely resulting from erroneous fungal fixing of the double stranded break, all resulting in a non-functional *pyrG*. Additionally, we did not find any off-target effects on the surrounding genes, highlighting the specificity of the RNP-based editing approach even in the absence of a provided fixing template. We thank the reviewer for this comment and feel that this additional information further strengthens the utility of our tools and underscores the utility of RNP-based editing in *A. oryzae* as a strategy for precise and efficient gene modification.

In addition, the *wA* mutants in NRRL strains were not investigated at a genetic level. It is not clear whether these mutants contain an expected deletion, which should be confirmed experimentally.

We thank the reviewer for this comment, and we regret that we did not include a detailed description for the generation of the *wA* mutants in the original manuscript. Based on the reviewer feedback, we have now included information about insertion strategy, integration efficiency, PCR confirmation, as well as protein expression phenotypes, for all *wA* mutants across the NRRL strain collection (Fig. S4). These data further support the conclusion that our toolkit can be used for genetic manipulation in diverse strains of *A. oryzae*.

4. In Figures 1D, S1F, and S4D, the graphs do not contain error bars, and the number of experimental trials was not written. As it is difficult to evaluate the reproducibility, these results do not seem to be enough to support the authors' conclusions.

We thank the reviewer for pointing out this unclear representation of the data. The reviewer is correct that the % depends on the number of screened transformants. Although the methodology is described in the methods section, we have updated the figures and the associated caption to better represent the data. Throughout the manuscript, a bar graph has been replaced with a table showing how many transformants were screened at condition and the frequency of transformants identified to harbor insertion at the locus of interest. This representation is directly comparable to how editing efficiency was presented in the previous publication of the plasmid-based CRISPR-Cas9 method for gene editing in *Aspergillus oryzae* (Katayama et al, 2019), as well as in other publications describing gene editing in filamentous fungi (Liu et al, 2015; Pohl et al, 2016). These results clearly highlight the efficiency of our method for gene editing across diverse loci in *A. oryzae*.

Similarly, in Lines 354-357 and Figure 4B, it seems to be difficult to conclude that the ergothioneine level of VMR-EG1-2 was higher than that of oyster mushroom without statistic tests.

We thank the reviewer for this comment. In response to the concern, we conducted a statistical test and we concluded that indeed the levels between oyster mushroom and VMR-EG1-2 are not statistically different (two-tailed t-test, $p > 0.05$). Therefore, we have revised the main text accordingly (lines 574-576).

5. The PCR results shown in figures are difficult to be precisely evaluated, especially > 4 kb, due to the gel with a high concentration of agarose. The expected band sizes were roughly described like ~ 4 kb and ~ 750 bp (Figures 1, S1, and S3). Furthermore, the strategies to construct *yA* and *niaD* mutants are not shown in Figure S3. The authors should show the construction strategies and describe the results precisely.

We thank the reviewer for pointing out this unclear representation of the data which made it difficult to interpret the PCR results characterizing insertion across diverse loci. For the *wA*, *niaD*, and *yA* loci, we have now changed the associated figures to represent more

clearly 1) the primers used 2) the predicted amplicon size 3) the agarose gel results. These changes can be found across Fig. 1 and Figs. S1, S4, S5, S6. This improved representation of the data further supports the utility and efficiency of our RNP-based method for gene integration and marker recycling in *A. oryzae*. We hope that this improved representation of the data will aid in replication and future design of constructs for integration and recycling across different loci by the research community.

6. GFP fluorescence was measured only with conidia in Figure 2. Considering the aim of this study, gene expression level in mycelia should be examined.

We disagree with the reviewer that the gene expression level in mycelia needs to be examined. The goal of the experiment presented in Figure 2 was to evaluate the strength of gene expression across different neutral loci. All loci harbored an integrated cassette driving GFP expression from the constitutive pTEF1 promoter, which was highly expressed in conidia. As all strains were evaluated with the same, strong constitutive promoter under the same morphological state, we feel that it is possible to assess the relative strength of protein expression using the experimental setup. This approach to comparing protein expression strength using flow cytometry in conidia has been demonstrated in *A. nidulans* (Wei et al, 2023). The neutral loci were used for engineering of ergothioneine and heme, demonstrating that they can be used for integration of overexpression of genes, which was the broader goal for identifying the neutral loci in the first place.

7. Only conidia formation was measured to show the fungal growth in Figure S6. To indicate the neutral loci, growth phenotypes should be shown.

We thank the reviewer for this comment. We have now added photos displaying colony morphology and growth phenotypes, which did not reveal any gross alterations in fungal physiology or growth in addition to the conidial yield reported in the original manuscript (Fig. S8). Additionally, as we did not observe any decrease in biomass yield between the heme strain (which utilized six different neutral loci) and the wild-type strain (Fig. S16), we are confident that there is a minimal effect of gene integration on fungal growth across these loci.

Minor points

Line 87: "1)" should be deleted.

This has been corrected.

Lines 113, 169, 268, 294, 315, and 422: The numbers indicating references should be upright (not italic).

This has been corrected.

Line 142: "locus" should be upright (not italic).

This has been corrected.

Lines 143, 148, 152, 521, and Figure S2: " Δ " should be upright (not italic).

This has been corrected

Lines 145-150: As "a diverse group of *A. oryzae*" is based on the whole-genome sequencing analyses, the description about whole-genome sequencing should be placed in the former sentence, and then the evaluation of the sequencing results should be described.

We thank the reviewer for this suggestion. The reviewer is correct that the diversity was established not only by the different geographical origins and industrial uses, but also by the phylogenetic analysis resulting from the whole genome sequencing. We have added a sentence to highlight the genome sequencing results and have added further analysis of the predicted number of coding sequences as well as the predicted natural products encoding capacity to highlight the differences and similarities between the strains (Table S2). We think this additional information will be of interest to the research community given the recent analyses of *Aspergillus* secondary metabolism. Additionally, we have revised the description of the sequencing and mutation of the NRRL strains in the main text (lines 194-226).

Lines 147 and 149-150: Table S1 > Table S2

This has been corrected

Line 165: Figure S1 > Figure S1F

This has been corrected

Lines 187-189: The conclusion here seems to be repeated with that mentioned in Lines 175-176. The authors should clarify these conclusions.

We thank the reviewer for pointing this out. We have removed the conclusion in Line 175-176 and now summarized only in the next paragraph.

Lines 189-191: It is not clear how this method "complements" the previous technique.

We thank the reviewer for pointing this out. We have edited the statement to differentiate our method relative to the previously published plasmid-based editing of *A. oryzae* (Katayama et al, 2019), and highlight its advantages throughout the manuscript (lines 147-159; lines 709-713).

Line 223: Figure 2C > Figure 2D. Then, no explanation about Figure 2C exists in the main text.

This has been corrected.

Line 223: The sentence "9 loci showed high-efficiency integration (>50%)" means that the 9 loci include *chro4_1* but not *chro3_1*. In the Figure 2 legend (Lines 542-543), however, the authors mentioned "all loci except for *chro4_1* displayed a high efficiency of integration (>50%)". Which is correct?

This was a writing mistake that has been corrected. We thank the reviewer for pointing this out.

Line 224 and Figure 2 legend: As names of the intergenic regions were in upright in Line 224, those in Figure 2 legend should be italic.

We have corrected the formatting according to this suggestion.

Lines 226 and 228: Figure 2D > Figure 2E

We have edited this.

Lines 227-228: *chro6-1* and *chro7-1* > *chro6_1* and *chro7_1* (not hyphen but underbar)

We have edited this.

Line 304: Why were the core promoters of *gpdA* and *hhfA* but not of *Ao_0583* used for construction of bidirectional promoter?

We selected *gpdA* and *hhfA* as a proof of concept that the bidirectional construction worked, and our data supports this conclusion.

Lines 304 and 567: *gpdA* and *hhfA* should be italic.

This has been corrected

Line 306: Figure 1F > Figure 3F

This has been corrected

Line 321: "*oryzae*" should be italic.

This has been corrected

Lines 351 and 583: "their own strong promoter" and "their own promoter" are confusing. "their own promoters" may mean those of *AO_Egt1* and *AO_Egt2* genes (but not *pTEF1*).

This has been corrected

Line 353: Does "co-expression" mean the expression from the bidirectional promoter? Dual gene integration also results in "co-expression".

This has been corrected

Line 386: It is not clear why Soy Leghemoglobin (= LegH?) was expressed.

The soy Leghemoglobin was chosen because that is the FDA-approved protein used as the heme source in IMPOSSIBLE meat, the leading plant-based meat incorporating heme

for flavor and color, as detailed in the main text (lines 626-630). As the safety and consumer acceptance has been established, we reasoned it would be a promising heme protein to express to potentially sequester free heme produced by overexpression of biosynthetic enzymes.

Supplementary Lines 214-215: In Table S11, why did the authors increase the LegH expression cassette by using the artificial promoters together with Bm3R1_VP16? Was the copy number increase effective for heme production?

Supplementary Line 182: Does leghemoglobin LegH really drive the hem biosynthesis flux?

We thank the reviewer for raising these questions. Heme biosynthesis in filamentous fungi is poorly characterized and has not been investigated beyond the deletion of a small number of individual biosynthetic enzymes in a limited range of hosts, making the engineering of heme biosynthesis a large challenge that had not been demonstrated in this group of organisms at the start of our study. It is known from yeasts that heme biosynthesis is a complex pathway regulated at transcriptional and post-translational level (Franken et al, 2011). Our initial engineering strategy is based on findings in *S. cerevisiae*, which have established that elevating production of both heme-binding proteins and biosynthetic enzymes is the key for increasing heme levels (Michener et al, 2012; Liu et al, 2014). The increase of both biosynthesis and a potential sink is thought to enhance heme levels by 1) creating an increased intracellular demand for heme, thus driving flux through the pathway and relieving potential feedback inhibition in biosynthetic enzymes, and 2) reducing the accumulation of free heme, which can be cytotoxic (Zoladek et al, 1996), as well as reducing the accumulation of toxic porphyrin intermediates. Based on these findings, we reasoned that by expressing the LegH under the strong promoter in the Synthetic Expression System (SES), in addition to under the significantly weaker promoter pTEF1, would similarly help alleviate potential intracellular stress caused by toxic heme or porphyrin elevation, at the same time as it could enhance pathway flux. We have highlighted this rationale in the manuscript (lines 662-682).

Line 406 and Figure 4: Figure 4E > Figure 4D

We have corrected this mistake.

Line 525: Unify the description of mCherry gene name. "mCherry"(Line 180) or "mcherry" (Line 525).

We have unified the names.

Line 782: What is the reference 64 "Administration, F. a. D. (2021)."?

We have edited the mistake in this reference, which was generated by the reference software used in the manuscript writing.

Supplementary Lines 16-19: In Fig. S1 legends, "D)" and "E)" should be reordered.

We have edited this Figure to more clearly capture the data.

Supplementary Lines 29 and 83: Ca9 > Cas9

We have corrected this mistake.

Supplementary Line 58: B) Successful > E) Successful

This has been corrected

Supplementary Lines 113, 240, 272, 304, 340, and 459: *Aspergillus oryzae* > A. *oryzae*

This has been corrected

Supplementary Line 128: "Candidate(s)" was repeated.

This has been corrected

Supplementary Lines 183-184: In Fig. S10 legend, the sentence here was just diverted from the main text (Lines 384-385).

This has been corrected

Supplementary Lines 207-209: In Table S9, clarify the upper and lower characters of primer sequences.

There was no functional meaning of upper vs lower case, it was a difference in formatting. To ensure consistency, all primers are now listed in upper case. Additionally, to facilitate interpretation and inform further use of primers, we have ensured that the names of primers correspond directly between experimental results presented in Figures and the names in Table S10 which contains all primers.

Supplementary Lines 207-209: In Table S9, not all the primers used in Figures 1 and S1 may be included.

We have edited the main text and supplementary materials to ensure consistency and clarity in which primers were used in which experiment.

Supplementary Lines 214-215: In the Description column of Table S11, the names of the ergothioneine biosynthesis genes EgtA and EgtB are different from those (AO_Egt1 and AO_Egt2) designated by the authors.

We have corrected this mistake in formatting.

Supplementary Line 226: Supplementary table S9 > Supplementary Table S9

This has been corrected

Supplementary Lines 246-247: grams > g, MgSO₄·7H₂O > MgSO₄·7H₂O (4 and 2 in subscript), KH₂PO₄ > KH₂PO₄ (2 and 4 in subscript)

This has been corrected. We are impressed with the reviewers attention to detail and grateful for all the corrections.

Supplementary Line 264: MgSO₄·7H₂O > MgSO₄·7H₂O

This has been corrected.

Supplementary Line 359: 1.2*10⁷ > 1.2*10⁷ (7 in superscript)

This has been corrected.

Supplementary Line 395: "1)" may not be necessary.

This has been corrected.

Supplementary Line 504: "genome" should be in upright (not italic).

This has been corrected.

Supplementary data S1: Although the occurrence values were shown, it is not clear how the highly expressed genes were determined. Was a threshold value of read counts determined?

We thank the reviewer for requesting clarification on this method. We explained the process of selection of the highly expressed genes in the methods section; however the reviewer's comment made us notice that this methods section was not very clear or readily accessible for all readers. To address this issue, we have rewritten this method in in the methods (lines 908-932) and added a brief statement in the legend of the Supplementary data S1 file plus a reference to these details in the supplementary info file.

Scale bars are missing in the microscopic images.

This has been added across all microscope images displaying fungal mycelia. We again thank the reviewer for the attention to detail, which has improved our manuscript.

References

Reider Apel A, d'Espaux L, Wehrs M, Sachs D, Li RA, Tong GJ, Garber M, Nnadi O, Zhuang W, Hillson NJ, Keasling JD, Mukhopadhyay A. A Cas9-based toolkit to program gene expression in *Saccharomyces cerevisiae*. *Nucleic Acids Res.* 2017 Jan 9;45(1):496-508. doi: 10.1093/nar/gkw1023. Epub 2016 Nov 28. PMID: 27899650; PMCID: PMC5224472.

Rantasalo A, Vitikainen M, Paasikallio T, Jäntti J, Landowski CP, Mojzita D. Novel genetic tools that enable highly pure protein production in *Trichoderma reesei*. *Sci Rep.* 2019 Mar 22;9(1):5032. doi: 10.1038/s41598-019-41573-8. PMID: 30902998; PMCID: PMC6430808.

Liu K, Sun B, You H, Tu JL, Yu X, Zhao P, Xu JW. Dual sgRNA-directed gene deletion in basidiomycete *Ganoderma lucidum* using the CRISPR/Cas9 system. *Microb Biotechnol.* 2020 Mar;13(2):386-396. doi: 10.1111/1751-7915.13534. Epub 2020 Jan 20. PMID: 31958883; PMCID: PMC7017817.

Gao D, Smith S, Spagnuolo M, Rodriguez G, Blenner M. Dual CRISPR-Cas9 Cleavage Mediated Gene Excision and Targeted Integration in *Yarrowia lipolytica*. *Biotechnol J*. 2018 Sep;13(9):e1700590. doi: 10.1002/biot.201700590. Epub 2018 Jun 11. PMID: 29809313.

Wei PL, Fan J, Yu J, Ma Z, Guo X, Keller NP, Li E, Lou C, Yin WB. Quantitative characterization of filamentous fungal promoters on a single-cell resolution to discover cryptic natural products. *Sci China Life Sci*. 2023 Apr;66(4):848-860. doi: 10.1007/s11427-022-2175-0. Epub 2022 Oct 21. PMID: 36287342.

Franken AC, Lokman BC, Ram AF, Punt PJ, van den Hondel CA, de Weert S. Heme biosynthesis and its regulation: towards understanding and improvement of heme biosynthesis in filamentous fungi. *Appl Microbiol Biotechnol*. 2011 Aug;91(3):447-60. doi: 10.1007/s00253-011-3391-3. Epub 2011 Jun 18. PMID: 21687966; PMCID: PMC3136693.

Michener JK, Nielsen J, Smolke CD. Identification and treatment of heme depletion attributed to overexpression of a lineage of evolved P450 monooxygenases. *Proc Natl Acad Sci U S A*. 2012 Nov 20;109(47):19504-9. doi: 10.1073/pnas.1212287109. Epub 2012 Nov 5. PMID: 23129650; PMCID: PMC3511110.

Liu L, Martínez JL, Liu Z, Petranovic D, Nielsen J. Balanced globin protein expression and heme biosynthesis improve production of human hemoglobin in *Saccharomyces cerevisiae*. *Metab Eng*. 2014 Jan;21:9-16. doi: 10.1016/j.ymben.2013.10.010. Epub 2013 Nov 2. PMID: 24188961.

Zoładek T, Nguyen BN, Rytka J. *Saccharomyces cerevisiae* mutants defective in heme biosynthesis as a tool for studying the mechanism of phototoxicity of porphyrins. *Photochem Photobiol*. 1996 Dec;64(6):957-62. doi: 10.1111/j.1751-1097.1996.tb01861.x. PMID: 8972638.

Reviewer #3 (Remarks to the Author):

L37, not sure about this statement, there are other technologies/platforms causing the same issues, and this statement is only blaming food industry. I would suggest that authors reformat the statement.

We thank the reviewer for raising this important point. We have edited the introduction accordingly to highlight that the food system is a major contributor to climate change, but not the only reason, which illustrates need for food innovation within the broader systemic challenge of improving planet health (lines 48-54).

Introduction: Could authors please add more information about 1- GMO, 2- Acceptability, 3-possible food safety concerns?

We thank the reviewer for raising the importance of discussing these timely and important topics. We have added a brief discussion of these points to the discussion, as we feel that section is where these topics are most appropriate and easily understood in light of our results (758-762).

Did author measure the color of the final product? I did not see the quality of the heme compared to the conventional heme in response to the thermal processing/cooking. The approach in this study is interesting, however, since the final product tends to be eaten, it should mimic the conventional meat. Could author also provide secondary structure information about the heme? This could be done by FTIR, and Chemometrics.

We thank the reviewer for recognizing the uniqueness of our approach and for raising this important point regarding comparison to actual foods. In addition to the development of new genetic tools for fungal engineering, our study represents a proof-of-concept of the application of bioengineering to alter the composition of edible fungal biomass, and we recognize that many additional steps are needed to translate our findings to actual products that are safe and palatable for human consumption. Our initial goal was to compare the levels of heme in the engineered strain to the leading plant-based product incorporating heme for flavor and color (IMPOSSIBLE foods), and these results highlighted that our engineered strain contains 40% of the levels found in IMPOSSIBLE. We feel that assessing the broader molecular and sensory similarity to conventional meat is beyond the scope of this study, including assessing the structure of heme during the cooking process, which undergoes many complex reactions upon cooking and heating, both in terms of its own breakdown, and its interactions with other small molecules and macromolecules inside the cell (Gandemer et al, 2020; Devaere et al, 2022; Zhang et al, 2022). We have clarified this limitation in the discussion section and highlighted the need for further work to compare the sensory properties and compositional similarity with actual food products as an important step towards translating our initial proof-of-concept into consumer-packaged goods (758-762).

However, in response to the reviewer comment, we have conducted further analysis of our high-resolution LC-MS data for heme and ergothioneine, which indicates that the

analyte between the standards, engineered strain, and foods used for comparison are identical, giving us further confidence that we have successfully engineered the levels of these flavor and health molecules in the *A. oryzae* biomass. This is an essential first step in the use of bioengineering to create actual food products that mimic actual foods, and we feel that the inclusion of these LC-MS data (Figs. S12 and S15) have further enhanced the conclusions and impact of our study.

References

Gandemer, G., Scislowski, V. , Portanguen, S. and Kondjoyan, A. (2020) The Impact of Cooking of Beef on the Supply of Heme and Non-Heme Iron for Humans. *Food and Nutrition Sciences*, **11**, 629-648. doi: [10.4236/fns.2020.117045](https://doi.org/10.4236/fns.2020.117045).

Devaere J, De Winne A, Dewulf L, Fraeye I, Šoljić I, Lauwers E, de Jong A, Sanctorem H. Improving the Aromatic Profile of Plant-Based Meat Alternatives: Effect of Myoglobin Addition on Volatiles. *Foods*. 2022 Jul 5;11(13):1985. doi: 10.3390/foods11131985. PMID: 35804800; PMCID: PMC9265346.

Yafei Zhang, Xiaojing Tian, Yuzhen Jiao, Yang Wang, Juan Dong, Ning Yang, Qinghua Yang, Wei Qu, Wenhong Wang. Free iron rather than heme iron mainly induces oxidation of lipids and proteins in meat cooking. *Food Chemistry*. Volume 382, 2022.

REVIEWER COMMENTS

Reviewer #1 (Remarks to the Author):

I have reviewed each comment raised in the first round of reviews, and the authors have satisfactorily addressed each individual point raised through the response document. I appreciate the added clarity on the engineering strategy and strain construction for heme and ergo strains, addition of protein content for both heme Fig. S13 and ergo strains S16 as well as additional details for LC-MS methodology (Fig S12 and S15).

Reviewer #2 (Remarks to the Author):

In the revised manuscript, most of the pointed issues have been addressed. The reviewer understands that the authors mainly focus on meat alternative application, but for synthetic biology genetic techniques developed in this study were not appropriately evaluated in the revised manuscript.

For instance, Fig. 4B indicates that the bidirectional promoter p4_2 is less active in ergothioneine production than pTEF1, which is recognized as a significantly weaker promoter by the authors (Lines 619-620). In addition, only one new promoter SES_Ao583 was used in heme production while using “weak” promoters of amyB and tef1 to express 5 of 6 genes as shown in Fig. S14 (however, many neutral loci were used effectively). All the newly developed promoters were shown to have much higher expression activities than known promoters in Fig. 3D. However, many of the promoters were not evaluated in the production experiments, thereby causing an uncertain validity of the genetic toolkit. The authors should weaken the descriptions for advantages in synthetic biology unless performing further experiments to show their practical uses.

High gene expression in mycelia is another important aspect for practical use. According to a less activity of bidirectional promoter and limited use of newly developed promoters, their expression levels might be decreased or lost in mycelia when comparing with those under known promoters. At least, the fluorescence intensities, which were analyzed only in conidia, should be confirmed with mycelia. A less activity and limited uses would not eliminate the possibility for different expression levels between conidia and mycelia.

These two issues should be addressed to appropriately claim a potential of synthetic biological applications with the developed genetic techniques.

In Lines 151-162 and 694-698 of the revised manuscript, the authors misleadingly mentioned the backgrounds of plasmid-mediated method. The reference 31 (Hao & Su, 2019) did employ an unusual method by combining constitutive expression Cas9 with in vitro transcribed gRNA, which resulted in unexpected insertions possibly by gRNA degradation or insufficient assembly of RNPs. Conclusions on off-target effects require a genome scale survey although the authors evaluated the effects only by analyzing the surrounding region of pyrG (Fig. S3E). The study in the reference 32 (Smith et al. 1991) regarding unintentional integration into the host genome did not use AMA1-based (autonomous

replication) plasmids, which are frequently used for genome editing in filamentous fungi. Furthermore, it seems that the sentence “the method requires laborious cloning...” improperly evaluates the plasmid-mediated method because authors’ method involves two sgRNAs preparation and complicated DNA construction with promoter, gene of interest, terminator, pyrG marker, and identical sequence of the flanking one.

Minor points

The unit of the scale bars should be μm but not μM , which is confusing with micro molar.

The descriptions of promoters should be unified. For example, the promoter of amyB gene was found as “PamyB” and “pAmyB”.

Line 202: Fig. S1 > Fig. S1A, B

Line 269: Fig. S1 > Fig. S1C, D

Line 544: “VMA-Eg_1_2” > “VMA-Eg1_2”

In Line 841, does “10 colonies” mean “10 colonies with white conidia formation”? If non-white colonies also appear, Fig. 1E results only represent the efficiency to detect expected DNA integration from white colonies.

Fig. S6B: niaD: RFP > niaD: mCherry

Fig. S6C: Labels of GFP and mCherry should be reversed.

Fig. S6E: niaD: gfp > niaD: mCherry

Table S4 (chro6_1): AO090038000287 exists between AO090038000288 and AO090038000286 in NCBI RefSeq assembly GCF_000184455.2 and CAoGD although the integrated site would not affect the conclusion.

Table S4: The locus ID of amyA is AO090003001591 but not AO090003001497. As the sgRNA sequences of amyA were designed in AO090003001591 although this would not affect the conclusion.

Fig. S14: “Heme-binding proteim” > “Heme-binding protein”

Supplementary Line 945: “then were then” > “were then”

Reviewer #1 (Remarks to the Author):

I have reviewed each comment raised in the first round of reviews, and the authors have satisfactorily addressed each individual point raised through the response document. I appreciate the added clarity on the engineering strategy and strain construction for heme and ergo strains, addition of protein content for both heme Fig. S13 and ergo strains S16 as well as additional details for LC-MS methodology (Fig S12 and S15).

We thank the reviewer for their helpful comments and suggestions, which improved the manuscript.

Reviewer #2 (Remarks to the Author):

In the revised manuscript, most of the pointed issues have been addressed. The reviewer understands that the authors mainly focus on meat alternative application, but for synthetic biology genetic techniques developed in this study were not appropriately evaluated in the revised manuscript.

For instance, Fig. 4B indicates that the bidirectional promoter p4_2 is less active in ergothioneine production than pTEF1, which is recognized as a significantly weaker promoter by the authors (Lines 619-620). In addition, only one new promoter SES_Ao583 was used in heme production while using “weak” promoters of amyB and tef1 to express 5 of 6 genes as shown in Fig. S14 (however, many neutral loci were used effectively). All the newly developed promoters were shown to have much higher expression activities than known promoters in Fig. 3D. However, many of the promoters were not evaluated in the production experiments, thereby causing an uncertain validity of the genetic toolkit. The authors should weaken the descriptions for advantages in synthetic biology unless performing further experiments to show their practical uses.

High gene expression in mycelia is another important aspect for practical use. According to a less activity of bidirectional promoter and limited use of newly developed promoters, their expression levels might be decreased or lost in mycelia when comparing with those under known promoters. At least, the fluorescence intensities, which were analyzed only in conidia, should be confirmed with mycelia. A less activity and limited uses would not eliminate the possibility for different expression levels between conidia and mycelia. These two issues should be addressed to appropriately claim a potential of synthetic biological applications with the developed genetic techniques.

We thank the reviewer for recognizing our experimental effort addressing the majority of the concerns in the previous revision and appreciate these thoughtful and detailed comments on the revised manuscript. We have addressed the remaining concerns in two ways and feel that the additional experiments and written clarification has significantly improved the manuscript.

-First, we have experimentally addressed the potential concern of conidial fluorescence not capturing whether there is also high protein expression in mycelia (Fig. S9). We have conducted an experiment using proteomics to validate the mycelial protein expression from a subset of the newly developed *A. oryzae* core promoters in the Synthetic Expression System. These data confirm that many of the new promoters are several-fold stronger than the commonly used pTEF1 and are useful for strong expression in mycelia. There was a significant correlation ($r=0.82$, $R^2=0.67$, $p<0.01$) between the flow cytometry and mycelium proteomics data. This supports our intended use of flow cytometry as an initial screening approach to

identify promising promoters. We have added a description of these data in the main text and added a caveat stating that following up on flow cytometry screening results in mycelia may be needed for establishing the precise promoter strength for submerged fermentations (main text, lines 494-511).

-Second, based on the reviewer feedback, we have toned down the language and significantly shortened the descriptions of practical applications for synthetic biology in the discussion section (main text, lines 752-773).

In Lines 151-162 and 694-698 of the revised manuscript, the authors misleadingly mentioned the backgrounds of plasmid-mediated method. The reference 31 (Hao & Su, 2019) did employ an unusual method by combining constitutive expression Cas9 with in vitro transcribed gRNA, which resulted in unexpected insertions possibly by gRNA degradation or insufficient assembly of RNPs. Conclusions on off-target effects require a genome scale survey although the authors evaluated the effects only by analyzing the surrounding region of pyrG (Fig. S3E). The study in the reference 32 (Smith et al. 1991) regarding unintentional integration into the host genome did not use AMA1-based (autonomous replication) plasmids, which are frequently used for genome editing in filamentous fungi. Furthermore, it seems that the sentence “the method requires laborious cloning...” improperly evaluates the plasmid-mediated method because authors’ method involves two sgRNAs preparation and complicated DNA construction with promoter, gene of interest, terminator, pyrG marker, and identical sequence of the flanking one.

-We thank the reviewer for this comment. We have revised our description of the plasmid-based editing method and the potential advantages of RNP complexes. We have toned down the language around specific examples and discussing potential drawbacks with constitutive Cas9/sgRNA expression and instead focused on highlighting our main reasons for developing the RNP-based approach for gene integration in *A. oryzae* (main text, lines 168-177).

Minor points

The unit of the scale bars should be μm but not μM , which is confusing with micro molar.

The descriptions of promoters should be unified. For example, the promoter of amyB gene was found as “PamyB” and “pAmyB”.

Line 202: Fig. S1 > Fig. S1A, B

Line 269: Fig. S1 > Fig. S1C, D

Line 544: “VMA-Eg_1_2” > “VMA-Eg1_2”

In Line 841, does “10 colonies” mean “10 colonies with white conidia formation”? If non-white colonies also appear, Fig. 1E results only represent the efficiency to detect expected DNA integration from white colonies.

Fig. S6B: niaD: RFP > niaD: mCherry

Fig. S6C: Labels of GFP and mCherry should be reversed.

Fig. S6E: niaD: gfp > niaD: mCherry

Table S4 (chro6_1): AO090038000287 exists between AO090038000288 and AO090038000286 in NCBI RefSeq assembly GCF_000184455.2 and CAoGD although the integrated site would not affect the conclusion.

Table S4: The locus ID of amyA is AO090003001591 but not AO090003001497. As the sgRNA sequences of amyA were designed in AO090003001591 although this would not affect the conclusion.

Fig. S14: “Heme-binding proteim” > “Heme-binding protein”

Supplementary Line 945: “then were then” > “were then”

We thank the reviewer for these detailed comments, which have been addressed and clarified throughout the manuscript.

REVIEWERS' COMMENTS

Reviewer #2 (Remarks to the Author):

The authors have improved the manuscript to address the concerns on their previous revisions, and the responses are satisfactory.